# GENERALIZED PRINCIPAL-AGENT PROBLEM WITH A LEARNING AGENT

**Tao Lin, Yiling Chen**
John A. Paulson School of Engineering and Applied Sciences
Harvard University
tlin@g.harvard.edu, yiling@seas.harvard.edu

## ABSTRACT

Generalized principal-agent problems, including Stackelberg games, contract design, and Bayesian persuasion, are a class of economic problems where an agent best responds to a principal's committed strategy. We study repeated generalized principal-agent problems under the assumption that the principal does not have commitment power and the agent uses algorithms to learn to respond to the principal. We reduce this problem to a one-shot generalized principal-agent problem where the agent approximately best responds. Using this reduction, we show that: (1) if the agent uses contextual no-regret learning algorithms with regret $\mathrm{Reg}(T)$, then the principal can guarantee utility at least $U^* - \Theta\big(\sqrt{\frac{\mathrm{Reg}(T)}{T}}\big)$, where $U^*$ is the principal's optimal utility in the classic model with a best-responding agent. (2) If the agent uses contextual no-swap-regret learning algorithms with swap-regret $\mathrm{SReg}(T)$, then the principal cannot obtain utility more than $U^* + O(\frac{\mathrm{SReg}(T)}{T})$. But (3) if the agent uses mean-based learning algorithms (which can be no-regret but not no-swap-regret), then the principal can sometimes do significantly better than $U^*$. These results not only refine previous results in Stackelberg games and contract design, but also lead to new results for Bayesian persuasion with a learning agent and all generalized principal-agent problems where the agent does not have private information.

## 1 INTRODUCTION

Classic economic models of principal-agent interactions, including auction design, contract design, and Bayesian persuasion, often assume that the agent is able to best respond to the strategy committed by the principal. For example, in Bayesian persuasion, the agent (receiver) needs to compute the posterior belief about the state of the world after receiving some information from the principal (sender) and take an optimal action based on the posterior belief; this requires the receiver accurately knowing the prior of the state as well as the signaling scheme used by the sender. In contract design, where a principal specifies an outcome-dependent payment scheme to incentivize the agent to take certain actions, the agent has to know the action-dependent outcome distribution in order to best respond to the contract. Requiring strong rationality assumptions, the best-responding behavior is often observed to be violated in practice (Camerer, 1998; Benjamin, 2019).

In this work, using Bayesian persuasion as the main example, we study general principal-agent problems under an alternative behavioral model for the agent: *learning*. The use of learning as a behavioral model dates back to early economic literature on learning in games (Brown, 1951; Fudenberg & Levine, 1998) and has been actively studied by computer scientists in recent years (e.g., Nekipelov et al. (2015); Braverman et al. (2018); Deng et al. (2019); Mansour et al. (2022); Cai et al. (2024); Guruganesh et al. (2024)). A learning agent no longer has perfect knowledge of the parameter of the game or the principal's strategy. Instead of best responding, which is no longer possible or well-defined, the agent chooses his action based on past interactions with the principal. We focus on *no-regret* learning, which requires the agent to not suffer a large average regret at the end of repeated interactions with the principal, for not taking the optimal action at hindsight. This is a mild requirement satisfied by many natural learning algorithms (e.g., $\varepsilon$-greedy, MWU, UCB, EXP-3) and can reasonably serve as a possible behavioral assumption for real-world agents.

Can the principal achieve a better outcome with a learning agent than with a best-responding agent? Previous works (Deng et al., 2019; Guruganesh et al., 2024) have shown that, in Stackelberg games and contract design, the leader/principal can obtain utility $U^* - o(1)$ against a no-regret learning follower/agent, where $U^*$ is the Stackelberg value, defined to be the principal's optimal utility in the classic model with a best-responding agent. On the other hand, if the agent does a stronger version of no-regret learning, called no-swap-regret learning (Hart & Mas-Colell, 2000; Blum & Mansour, 2007), then the principal cannot obtain utility more than the Stackelberg value $U^* + o(1)$. Interestingly, the conclusion that no-swap-regret learning can cap the principal's utility at $U^* + o(1)$ does not hold when the agent has private information, such as in auctions (Braverman et al., 2018) and Bayesian Stackelberg games (Mansour et al., 2022): the principal can sometimes exploit a no-swap-regret learning agent with private information to do much better than $U^*$ in those games.

Three natural questions then arise: (1) What is the largest class of principal-agent problems under which the agent's no-swap-regret learning can cap the principal's utility at the Stackelberg value $U^* + o(1)$? (2) In cases where the principal's optimal utility against a learning agent is bounded by $[U^* - o(1), U^* + o(1)]$, what is the exact magnitude of the $o(1)$ terms? (3) Instead of analyzing games like Stackelberg games and contract design separately, can we analyze all principal-agent problems with learning agents in a unified way?

**Our contributions.** Our work defines a general model of principal-agent problems with a learning agent, answering all questions (1) - (3). For (1), we show that the principal's utility is bounded around $U^*$ in all generalized principal-agent problems where the agent does not have private information but the principal can be privately informed. In particular, this includes complete-information games like Stackelberg games and contract design, as well as Bayesian persuasion where the sender/principal privately observes the state of the world.

For (2) and (3), we provide a unified analytical framework to derive tight bounds on the principal's achievable utility against a no-regret or no-swap-regret learning agent in all generalized principal-agent problems where the agent does not have private information. Specifically, we explicitly characterize the $o(1)$ difference between the principal's utility and $U^*$ in terms of the agent's regret.

**Result 1** (from Theorems 3.1, 4.1, 4.2). *Against a no-regret learning agent with regret $\mathrm{Reg}(T)$ in $T$ periods, the principal can obtain an average utility of at least $U^* - O\big(\sqrt{\frac{\mathrm{Reg}(T)}{T}}\big)$.*

**Result 2** (from Theorems 3.4, 4.1, 4.2). *Against a no-swap-regret learning agent with swap-regret $\mathrm{SReg}(T)$ in $T$ periods, the principal cannot obtain average utility larger than $U^* + O\big(\frac{\mathrm{SReg}(T)}{T}\big)$.*

Interestingly, the squared root bound $U^* - O\big(\sqrt{\frac{\mathrm{Reg}(T)}{T}}\big)$ in Result 1 and the linear bound $U^* + O\big(\frac{\mathrm{SReg}(T)}{T}\big)$ in Result 2 are not symmetric. We show that such an asymmetry is intrinsic: there exist cases where the principal cannot achieve better than $U^* - O\big(\sqrt{\frac{\mathrm{Reg}(T)}{T}}\big)$ utility.

**Result 3** (from Theorem 3.3 and Example 4.1). *There is a Bayesian persuasion instance where, for any strategy of the principal, there is a no-swap-regret learning algorithm for the agent under which the principal's utility is at most $U^* - \Omega\big(\sqrt{\frac{\mathrm{SReg}(T)}{T}}\big)$. The same holds for no-regret algorithms.*

Results 1, 2, 3 together characterize the range of utility achievable by the principal against a no-swap-regret learning agent: $[U^* - \Theta\big(\sqrt{\frac{\mathrm{SReg}(T)}{T}}\big), U^* + O\big(\frac{\mathrm{SReg}(T)}{T}\big)]$. For no-regret but not necessarily no-swap-regret algorithms, the upper bound $U^* + O\big(\frac{\mathrm{Reg}(T)}{T}\big)$ does not hold:

**Result 4** (Theorem 3.5). *We construct a Bayesian persuasion instance where, against a no-regret but not no-swap-regret learning agent (in particular, mean-based learning agent), the principal can do significantly better than the Stackelberg value $U^*$.*

In summary, our Results 1, 2, 3 not only refine previous works on playing against learning agents in specific games by characterizing the principal's utility exactly, but also generalize to all principal-agent problems where the agent does not have private information. In particular, when applied to Bayesian persuasion, our results imply that the sender cannot exploit a no-swap-regret learning receiver even if the sender possesses informational advantage over the receiver.

**Some intuition.** What is the intuition behind the asymmetry between the worst-case utility $U^* - \Theta(\sqrt{\frac{\text{SReg}(T)}{T}})$ and the best-case utility $U^* + O(\frac{\text{SReg}(T)}{T})$ that the principal can obtain against a no-swap-regret learning agent? At a high level, a no-swap-regret learning agent is *approximately best responding* to the principal's average strategy over all $T$ periods, with the degree of approximate best response measured by the average regret $\frac{\text{SReg}(T)}{T} = \delta$. However, because no-swap-regret learning algorithms are randomized, they correspond to randomized approximately best responding strategies of the agent that are worse than the best responding strategy by a margin of $\delta$ *in expectation*. This means that the agent might take $\sqrt{\delta}$-sub-optimal actions with probability $\sqrt{\delta}$, which can cause a loss of 1 to the principal's utility with probability $\sqrt{\delta}$. So, the principal's expected utility can be decreased to $U^* - \Omega(\sqrt{\delta}) = U^* - \Omega(\sqrt{\frac{\text{SReg}(T)}{T}})$ in the worst case. On the other hand, when considering the principal's best-case utility, we care about the $\delta$-approximately-best-responding strategy of the agent that maximizes the principal's utility. That strategy turns out to be equivalent to a deterministic strategy that gives the principal a utility of at most $U^* + O(\delta) = U^* + O(\frac{\text{SReg}(T)}{T})$. This explains the asymmetry between the worst-case and best-case bounds.

# 2 GENERALIZED PRINCIPAL-AGENT PROBLEM WITH A LEARNING AGENT

This section defines our model, *generalized principal-agent problem with a learning agent*. This model includes Stackelberg games, contract design, and Bayesian persuasion with learning agents.

## 2.1 GENERALIZED PRINCIPAL-AGENT PROBLEM

*Generalized principal-agent problem*, proposed by Myerson (1982); Gan et al. (2024), is a general model that includes auction design, contract design, Stackelberg games, and Bayesian persuasion. While Myerson (1982) and Gan et al. (2024) allow the agent to have private information, our model assumes an agent with no private information. There are two players in a generalized principal-agent problem: a principal and an agent. The principal has a convex, compact decision space $\mathcal{X}$ and the agent has a finite action set $A$. The principal and the agent have utility functions $u, v : \mathcal{X} \times A \to \mathbb{R}$. We assume that $u(x, a)$, $v(x, a)$ are linear in $x \in \mathcal{X}$, which is satisfied by all the examples of generalized principal-agent problems we will consider (Bayesian persuasion, Stackelberg games, contract design). There is a signal/message set $S$. Signals are usually interpreted as recommendations of actions for the agent, where $S = A$, but we allow any signal set of size $|S| \geq |A|$. A strategy of the principal is a distribution $\pi \in \Delta(\mathcal{X} \times S)$ over pairs of decision and signal. When the utility functions $u, v$ are linear, it is without loss of generality to assume that the principal does not randomize over multiple decisions for one signal (Gan et al., 2024), namely, the principal chooses a distribution over signals and a unique decision $x_s$ associated with each signal $s \in S$. So, we can write a principal strategy as $\pi = \{(\pi_s, x_s)\}_{s \in S}$ where $\pi_s \geq 0$ is the probability of signal $s \in S$, $\sum_{s \in S} \pi_s = 1$, and $x_s \in \mathcal{X}$. There are two variants of generalized principal-agent problems:

- *Unconstrained* (Myerson, 1982): there is no restriction on the principal's strategy $\pi$.
- *Constrained* (Gan et al., 2024): the principal's strategy $\pi$ has to satisfy constraint $\sum_{s \in S} \pi_s x_s \in \mathcal{C}$ where $\mathcal{C} \subseteq \mathcal{X}$ is some convex set.

Unconstrained generalized principal-agent problems include contract design and Stackelberg games. Constrained generalized principal-agent problems include Bayesian persuasion (see Section 2.3).

In a one-shot generalized principal-agent problem where the principal has commitment power, the principal first commits to a strategy $\pi = \{(\pi_s, x_s)\}_{s \in S}$, then nature draws a signal $s \in S$ according to the distribution $\{\pi_s\}_{s \in S}$ and sends $s$ to the agent (note: due to the commitment assumption, this is equivalent to revealing the pair $(s, x_s)$ to the agent), then the agent takes an action $a_s \in \arg\max_{a \in A} v(x_s, a)$ that maximizes its utility, and the principal obtains utility $u(x_s, a_s)$. The principal aims to maximize its expected utility $\mathbb{E}_{s \sim \pi}[u(x_s, a_s)]$ by choosing the strategy $\pi$.

## 2.2 LEARNING AGENT

Now we define the model of generalized principal-agent problem with a learning agent. The game is repeated for $T$ rounds. Unlike the static model above, the principal here does not commit. The agent

does not know the strategy $\pi^t$ or the decision $x^t$ of the principal at each round. Instead, the agent uses some adaptive algorithm to learn which action to take in response to each possible signal.

---

**Generalized Principal-Agent Problem with a Learning Agent**

In each round $t = 1, \ldots, T$:

(1) Using some algorithm that learns from history (including signals, actions, and utility feedback in the past, described in details later), the agent chooses a strategy $\rho^t : S \to \Delta(A)$ that maps each possible signal $s \in S$ to a distribution over actions $\rho^t(s) \in \Delta(A)$.

(2) The principal chooses a strategy $\pi^t = \{(\pi^t_s, x^t_s)\}_{s \in S}$, which is a distribution over signals $S$ and a decision $x^t_s \in \mathcal{X}$ associated with each signal.

(3) Nature draws signal $s^t \sim \pi^t$ and reveals it. The principal makes decision $x^t = x^t_{s^t}$. The agent draws action $a^t \sim \rho^t(s^t)$.

(4) The principal and the agent obtain utility $u^t = u(x^t, a^t)$ and $v^t = v(x^t, a^t)$. The agent observes some feedback (e.g., $v^t(x^t, a^t)$ or $x^t$).

---

We assume that the principal knows the utility functions $u$ and $v$, has some knowledge about the agent's learning algorithm, and aims to maximize the expected average utility $\frac{1}{T}\mathbb{E}\big[\sum_{t=1}^T u(x^t, a^t)\big]$.

**Agent's learning problem.** The agent's learning problem can be regarded as a *contextual multi-armed bandit problem* (Tyler Lu et al., 2010) where $A$ is the set of arms, and a signal $s^t \in S$ serves as a context that affects the utility of each arm $a \in A$. The agent picks an arm to pull based on the current context $s^t$ and the historical information about each arm under different contexts, adjusting its strategy over time based on the feedback collected after each round.

What feedback can the agent observe after each round? One may assume that the agent sees the principal's decision $x^t$ after each round (this is call *full-information* feedback in the multi-armed bandit literature), or the utility $v^t = v(x^t, a^t)$ obtained in that round (this is called *bandit feedback*), or some unbiased estimate of $v(x^t, a^t)$. We do not make specific assumptions on the feedback. All we need is that the feedback is sufficient for the agent to achieve contextual no-regret or contextual no-swap-regret, which are defined below:

**Definition 2.1.** *The agent's learning algorithm is said to satisfy:*

- contextual no-regret *if: there is a function $\mathrm{CReg}(T) = o(T)$ such that, for any strategy of the principal, for any deviation function $d : S \to A$, the regret of the agent not deviating according to $d$ is at most $\mathrm{CReg}(T)$: $\mathbb{E}\big[\sum_{t=1}^T \big(v(x^t, d(s^t)) - v(x^t, a^t)\big)\big] \leq \mathrm{CReg}(T)$.*

- contextual no-swap-regret *if: there is a function $\mathrm{CSReg}(T) = o(T)$ such that, for any strategy of the principal, for any deviation function $d : S \times A \to A$, the regret of the receiver not deviating according to $d$ is at most $\mathrm{CSReg}(T)$: $\mathbb{E}\big[\sum_{t=1}^T \big(v(x^t, d(s^t, a^t)) - v(x^t, a^t)\big)\big] \leq \mathrm{CSReg}(T)$.*

*We call $\mathrm{CReg}(T)$ and $\mathrm{CSReg}(T)$ the* contextual regret *and* contextual swap-regret *of the agent.*

Contextual no-regret is implied by contextual no-swap-regret because the latter has a larger set of deviation functions. Contextual no-(swap-)regret algorithms with $O(|A|\sqrt{|S|T})$ contextual (swap-)regret are known to exist under bandit feedback. In fact, they can be easily constructed by running an ordinary no-(swap-)regret algorithm for each context independently. See Appendix B for details.

## 2.3 SPECIAL CASE: BAYESIAN PERSUASION WITH A LEARNING AGENT

We show that *Bayesian persuasion* (Kamenica & Gentzkow, 2011) is a special case of constrained generalized principal-agent problems. We will also show that Bayesian persuasion is in fact equivalent to *cheap talk* (Crawford & Sobel, 1982) under our learning agent model.

**Bayesian persuasion as a generalized principal-agent problem.** There are two players in Bayesian persuasion: a sender (principal) and a receiver (agent). There are a finite set $\Omega$ of states of the world, a signal set $S$, an action set $A$, a prior distribution $\mu_0 \in \Delta(\Omega)$ over the states, and utility functions $u, v : \Omega \times A \to \mathbb{R}$ for the sender and the receiver. When the state is $\omega \in \Omega$ and the receiver takes action $a \in A$, the sender and the receiver obtain utility $u(\omega, a)$, $v(\omega, a)$, respectively.

Both players know $\mu_0$, but only the sender has access to the realized state $\omega \sim \mu_0$. The sender commits to some signaling scheme $\pi : \Omega \to \Delta(S)$, mapping any state to a probability distribution over signals, to partially reveal information about the state $w$ to the receiver. In the classic model, after receiving a signal $s \in S$, the receiver will form the posterior belief $\mu_s \in \Delta(\Omega)$ about the state: $\mu_s(\omega) = \frac{\mu_0(\omega)\pi(s|\omega)}{\pi_s}$, where $\pi_s = \sum_{\omega \in \Omega} \mu_0(\omega)\pi(s|\omega)$ is the total probability that signal $s$ is sent, and take an optimal action with respect to $\mu_s$, i.e., $a_s \in \arg\max_{a \in A} \sum_{\omega \in \Omega} \mu_s(\omega)v(\omega, a)$. The sender aims to find a signaling scheme to maximizde its expected utility $\mathbb{E}[u(\omega, a_s)]$.

It is well-known (Kamenica & Gentzkow, 2011) that a signaling scheme $\pi : \Omega \to \Delta(S)$ decomposes the prior $\mu_0$ into a distribution over posteriors whose average is equal to the prior $\mu_0$:

$$\sum_{s \in S} \pi_s \mu_s = \mu_0 \in \{\mu_0\} =: \mathcal{C}, \quad \sum_{s \in S} \pi_s = 1. \tag{1}$$

Equation (1) is called the *Bayes plausibility* condition. Conversely, any distribution over posteriors $\{(p_s, \mu_s)\}_{s \in S}$ satisfying Bayes plausibility $\sum_{s \in S} p_s \mu_s = \mu_0$ can be converted into a signaling scheme that sends signal $s$ with probability $p_s$. Thus, we can use a distribution over posteriors $\{(\pi_s, \mu_s)\}_{s \in S}$ satisfying Bayes plausibility to represent a signaling scheme. Then, let's equate the posterior belief $\mu_s$ in Bayesian persuasion to the principal's decision $x_s$ in the generalized principal-agent problem, so the principal/sender's decision space becomes $\mathcal{X} = \Delta(\Omega)$. The Bayes plausibility condition (1) becomes the constraint in the constrained generalized principal-agent problem. When the agent/receiver takes action $a$, the principal/sender's (expected) utility under decision/posterior $x_s = \mu_s$ is $u(x_s, a) = \mathbb{E}_{\omega \sim \mu_s} u(\omega, a) = \sum_{\omega \in \Omega} \mu_s(\omega)u(\omega, a)$. Suppose the agent takes action $a_s$ given signal $s \in S$. Then we see that the sender's utility of using signaling scheme $\pi$ in Bayesian persuasion (left) is equal to the principal's utility of using strategy $\pi$ in the generalized principal-agent problem (right):

$$\sum_{\omega \in \Omega} \mu_0(\omega) \sum_{s \in S} \pi(s|\omega)u(\omega, a_s) = \sum_{s \in S} \pi_s \sum_{\omega \in \Omega} \mu_s(\omega)u(\omega, a_s) = \sum_{s \in S} \pi_s u(x_s, a_s) = \mathbb{E}_{s \sim \pi}[u(x_s, a)].$$

Similarly, the agent/receiver's utilities in the two problems are equal. The utility functions $u(x, a)$, $v(x, a)$ are linear in the principal's decision $x \in \mathcal{X}$, satisfying our assumption.

**Persuasion (or cheap talk) with a learning agent**     When specialized to Bayesian persuasion, the generalized principal-agent problem with a learning agent becomes the following:

---

**Persuasion (or Cheap Talk) with a Learning Receiver**

In each round $t = 1, \ldots, T$, the following events happen:

(1) Using some algorithm that learns from history, the receiver chooses a strategy $\rho^t : S \to \Delta(A)$ that maps each signal $s \in S$ to a distribution over actions $\rho^t(s) \in \Delta(A)$.

(2) The sender chooses a signaling scheme $\pi^t : \Omega \to \Delta(S)$.

(3) A state of the world $\omega^t \sim \mu_0$ is realized, observed by the sender but not the receiver. The sender sends signal $s^t \sim \pi^t(\omega^t)$ to the receiver. The receiver draws action $a^t \sim \rho^t(s)$.

(4) The sender obtains utility $u^t = u(\omega^t, a^t)$ and the receiver obtains utility $v^t = v(\omega^t, a^t)$.[a]

---

[a] The definition of utility here, $u(\omega^t, a^t), v(\omega^t, a^t)$, is different from the definition in the general model, which was the expected utility on decision/posterior $x^t$, $u(x^t, a^t), v(x^t, a^t)$. Because we only care about the sender's expected utility and the receiver's expected regret, this difference does not matter.

---

In the above model, the receiver does not need to know the prior $\mu_0$ or the sender's signaling scheme because his multi-armed bandit learning algorithm does not need such information. This makes the receiver's and the sender's decisions simultaneous, which corresponds to the *cheap talk* model (Crawford & Sobel, 1982) where the sender does commit to the signaling scheme. So, our "persuasion with a learning receiver" model is equivalent to "cheap talk with a learning receiver".

## 3    REDUCTION FROM LEARNING TO APPROXIMATE BEST RESPONSE

In this section, we reduce the generalized principal-agent problem with a learning agent to the problem with an approximately-best-responding agent. We show that, if the agent uses contextual no-regret learning algorithms, then the principal can obtain an average utility that is at least the

"maxmin" approximate-best-response objective $\underline{\text{OBJ}}^{\mathcal{R}}\big(\text{CReg}(T)/T\big)$ (to be defined below). On the other hand, if the agent does contextual no-swap-regret learning, then the principal cannot do better than the "maxmax" approximate-best-response objective $\overline{\text{OBJ}}^{\mathcal{R}}\big(\text{CSReg}(T)/T\big)$. In addition, if the agent uses some learning algorithms that are no-regret but not no-swap-regret, the principal can sometimes do better than the "maxmax" objective $\overline{\text{OBJ}}^{\mathcal{R}}\big(\text{CSReg}(T)/T\big)$.

## 3.1 Generalized Principal-Agent Problem with Approximate Best Response

We first define the generalized principal-agent problem with an *approximately-best-responding* agent. The classic generalized principal-agent problem (Section 2.1) assumes that, after receiving a signal $s \in S$ (and observing the principal's decision $x_s \in \mathcal{X}$), the agent will take an optimal action with respect to $x_s$. This means that the agent uses a strategy $\rho^*$ that *best responds* to the principal's strategy $\pi$:

$$\rho^*(s) \in \arg\max_{a \in A} v(x_s, a), \ \forall s \in S \quad \Longrightarrow \quad \rho^* \in \arg\max_{\rho: S \to \Delta(A)} V(\pi, \rho). \tag{2}$$

Here, $V(\pi, \rho) = \sum_{s \in S} \pi_s \sum_{a \in A} \rho(a|s) v(x_s, a)$ denotes the expected utility of the agent when the principal uses strategy $\pi$ and the agent uses (randomized) strategy $\rho: S \to \Delta(A)$.

Here, we allow the agent to *approximately* best respond. Let $\delta \geq 0$ be a parameter. We define two types of $\delta$-best-responding strategies for the agent: deterministic and randomized.

- A deterministic strategy $\rho$: for each signal $s \in S$, the agent takes an action $a$ that is $\delta$-optimal for $x_s$. Denote this set of strategies by $\mathcal{D}_\delta(\pi)$:

$$\mathcal{D}_\delta(\pi) = \big\{ \rho: S \to A \mid v(x_s, \rho(s)) \geq v(x_s, a') - \delta, \ \forall a' \in A \big\}. \tag{3}$$

- A randomized strategy $\rho$: for each signals $s$, the agent can take a randomized action. The expected utility of $\rho$ is at most $\delta$-worst than the best strategy $\rho^*$.

$$\mathcal{R}_\delta(\pi) = \big\{ \rho: S \to \Delta(A) \mid V(\pi, \rho) \geq V(\pi, \rho^*) - \delta \big\}. \tag{4}$$

Equivalently, $\mathcal{R}_\delta(\pi) = \big\{ \rho: S \to \Delta(A) \mid V(\pi, \rho) \geq V(\pi, \rho') - \delta, \ \forall \rho': S \to A \big\}$.

Our model of approximately-best-responding agent includes, for example, two other models in the Bayesian persuasion literature that also relax the agent's Bayesian rationality assumption: the quantal response model (proposed by (McKelvey & Palfrey, 1995) in normal-form games and studied by (Feng et al., 2024) in Bayesian persuasion) and a model where the agent makes mistakes in Bayesian update (de Clippel & Zhang, 2022). See Appendix C for details.

**Principal's objectives.** With an approximately-best-responding agent, we will study two types of objectives for the principal. The first type is the maximal utility that the principal can obtain if the agent approximately best responds in the *worst* way for the principal: for $X \in \{\mathcal{D}, \mathcal{R}\}$, define

$$\underline{\text{OBJ}}^X(\delta) = \sup_\pi \min_{\rho \in X_\delta(\pi)} U(\pi, \rho), \tag{5}$$

where $U(\pi, \rho) = \sum_{s \in S} \pi_s \sum_{a \in A} \rho(a|s) u(x_s, a)$ is the principal's expected utility when the principal uses strategy $\pi$ and the agent uses strategy $\rho$. We used "sup" in (5) because the maximizer does not necessarily exist. $\underline{\text{OBJ}}^X(\delta)$ is a "maxmin" objective and can be regarded as the objective of a "robust generalized principal-agent problem".

The second type of objectives is the maximal utility that the principal can obtain if the agent approximately best responds in the *best* way:

$$\overline{\text{OBJ}}^X(\delta) = \max_\pi \max_{\rho \in X_\delta(\pi)} U(\pi, \rho). \tag{6}$$

This is a "maxmax" objective that quantifies the maximal extent to which the principal can exploit the agent's irrational behavior.

Clearly, $\underline{\text{OBJ}}^X(\delta) \leq \underline{\text{OBJ}}^X(0) \leq \overline{\text{OBJ}}^X(0) \leq \overline{\text{OBJ}}^X(\delta)$. And we note that $\overline{\text{OBJ}}^X(0) = \overline{\text{OBJ}}(0)$ is independent of $X$ and equal to the optimal utility of the principal in the classic generalized principal-agent problem, which we denote by $U^*$:

$$U^* = \overline{\text{OBJ}}(0) = \max_\pi \max_{\rho: \text{ best-response to } \pi} U(\pi, \rho). \tag{7}$$

Finally, we note that, because $\mathcal{D}_0(\pi) \subseteq \mathcal{D}_\delta(\pi) \subseteq \mathcal{R}_\delta(\pi)$, the chain of inequalities $\underline{\text{OBJ}}^{\mathcal{R}}(\delta) \leq \underline{\text{OBJ}}^{\mathcal{D}}(\delta) \leq U^* \leq \overline{\text{OBJ}}^{\mathcal{D}}(\delta) \leq \overline{\text{OBJ}}^{\mathcal{R}}(\delta)$ hold.

## 3.2 Agent's No-Regret Learning: Lower Bound on Principal's Utility

**Theorem 3.1.** *Suppose the agent uses a contextual no-regret learning algorithm with a contextual regret upper bounded by* $\text{CReg}(T)$. *The principal knows* $\text{CReg}(T)$ *but not the exact learning algorithm of the agent. By using some fixed strategy* $\pi^t = \pi$ *in all* $T$ *rounds, the principal can obtain an average utility* $\frac{1}{T}\mathbb{E}\big[\sum_{t=1}^T u(x^t, a^t)\big]$ *that is arbitrarily close to* $\underline{\text{OBJ}}^{\mathcal{R}}\big(\frac{\text{CReg}(T)}{T}\big)$.

To prove Theorem 3.1, we provide a lemma (with proof in Appendix E.1) to relate the agent's regret and the principal's utility in the learning model to those in the static model. We define some notations. Let the principal use some fixed strategy $\pi^t = \pi$ and the agent use some learning algorithm. Let $p^t_{a|s} = \Pr[a^t = a \mid s^t = s]$ be the probability that the agent's algorithm chooses action $a$ conditioning on signal $s$ being sent in round $t$. Let $\rho : S \to \Delta(A)$ be a randomized agent strategy that, given signal $s$, chooses each action $a \in A$ with probability $\rho(a|s) = \frac{\sum_{t=1}^T p^t_{a|s}}{T}$.

**Lemma 3.2.** *When the principal uses a fixed strategy* $\pi^t = \pi$ *in all* $T$ *rounds, the regret of the agent not deviating according to* $d : S \to A$ *is equal to* $\frac{1}{T}\mathbb{E}\big[\sum_{t=1}^T \big(v(x^t, d(s^t)) - v(x^t, a^t)\big)\big] = V(\pi, d) - V(\pi, \rho)$, *and the average utility of the principal* $\frac{1}{T}\mathbb{E}\big[\sum_{t=1}^T u(x^t, a^t)\big]$ *is equal to* $U(\pi, \rho)$.

*Proof of Theorem 3.1.* By Lemma 3.2 and the no-regret condition that the agent's regret $\mathbb{E}\big[\sum_{t=1}^T \big(v(x^t, d(s^t)) - v(x^t, a^t)\big)\big] \leq \text{CReg}(T)$, we have

$$V(\pi, d) - V(\pi, \rho) = \frac{1}{T}\mathbb{E}\Big[\sum_{t=1}^T \Big(v(x^t, d(s^t)) - v(x^t, a^t)\Big)\Big] \leq \frac{\text{CReg}(T)}{T}, \quad \forall d : S \to A.$$

This means that the agent's randomized strategy $\rho$ is a $\delta = \frac{\text{CReg}(T)}{T}$-best-response to the principal's fixed signaling scheme $\pi$, $\rho \in \mathcal{R}_{\delta = \frac{\text{CReg}(T)}{T}}(\pi)$. This holds for any $\pi$. In particular, if for any $\varepsilon > 0$ the principal uses a signaling scheme $\pi^\varepsilon$ that obtains an objective that is $\varepsilon$-close to $\underline{\text{OBJ}}^{\mathcal{R}}(\delta) = \sup_\pi \min_{\rho \in \mathcal{R}_\delta(\pi)} U(\pi, \rho)$, then the principal obtains an expected utility of, by Lemma 3.2,

$$\frac{1}{T}\mathbb{E}\Big[\sum_{t=1}^T u(a^t, \omega^t)\Big] = U(\pi^\varepsilon, \rho) \geq \min_{\rho \in \mathcal{R}_\delta(\pi^\varepsilon)} U(\pi^\varepsilon, \rho) \geq \underline{\text{OBJ}}^{\mathcal{R}}\Big(\delta = \frac{\text{CReg}(T)}{T}\Big) - \varepsilon$$

in the learning model. Letting $\varepsilon \to 0$ proves the theorem. $\qquad\square$

We then show that the result in Theorem 3.1 is tight: there exist cases where the principal cannot do better than $\underline{\text{OBJ}}^{\mathcal{R}}\big(\frac{\text{CReg}(T)}{2T}\big)$ even using adaptive strategies (see Appendix E.2 for the proof):

**Theorem 3.3.** *For any adaptive strategy of the principal, there exists a contextual no-regret learning algorithm for the agent under which the principal's average utility is no more than* $\underline{\text{OBJ}}^{\mathcal{R}}\big(\frac{\text{CReg}(T)}{2T}\big)$. *There also exists a contextual no-swap-regret learning algorithm for the agent under which the principal's average utility is no more than* $\underline{\text{OBJ}}^{\mathcal{R}}\big(\frac{\text{CSReg}(T)}{2T}\big)$.

## 3.3 Agent's No-Swap-Regret Learning: Upper Bound on Principal's Utility

**Theorem 3.4.** *Against a contextual no-swap-regret learning agent, the principal cannot obtain utility more than* $\frac{1}{T}\mathbb{E}\big[\sum_{t=1}^T u(x^t, a^t)\big] \leq \overline{\text{OBJ}}^{\mathcal{R}}\big(\frac{\text{CSReg}(T)}{T}\big)$ *even using adaptive strategies.*

The key idea to prove this theorem is to think of the signal $s^t \sim \pi^t$ from the principal and the action $a^t \sim \rho^t(s^t)$ recommended by the agent's learning algorithm together as a joint signal $(s^t, a^t)$ from some hypothetical signaling scheme $\pi'$. The agent takes the recommended action $a^t$, namely using the mapping $(s^t, a^t) \mapsto a^t$ as his strategy, in response to $\pi'$. A no-swap-regret algorithm guarantees that the agent is at most $\frac{\text{CSReg}(T)}{T}$ worse compared to the best-responding strategy $d^* : S \times A \to A$. So, the agent's overall strategy is a $\frac{\text{CSReg}(T)}{T}$-approximate best response to $\pi'$, which limits the principal's overall utility to be at most $\overline{\text{OBJ}}^{\mathcal{R}}\big(\frac{\text{CSReg}(T)}{T}\big)$. See details in Apendix E.3.

### 3.4 AGENT'S MEAN-BASED LEARNING: EXPLOITABLE BY THE PRINCIPAL

Many no-regret (but not no-swap-regret) learning algorithms (e.g., MWU, FTPL, EXP-3) satisfy the following *contextual mean-based* property:

**Definition 3.1** (Braverman et al. (2018)). *Let $\sigma_s^t(a) = \sum_{j \in [t]:s^j=s} v(\omega^j, a)$ be the sum of historical utilities of the receiver in the first $t$ rounds if he takes action $a$ when the signal/context is $s$. An algorithm is called $\gamma$-mean-based if: whenever $\exists a'$ such that $\sigma_s^{t-1}(a) < \sigma_s^{t-1}(a') - \gamma T$, the probability that the algorithm chooses action $a$ at round $t$ if the context is $s$ is $\Pr[a^t = a|s^t = s] < \gamma$, with $\gamma = o(1)$.*

**Theorem 3.5.** *There exists a Bayesian persuasion instance where, as long as the receiver does $\gamma$-mean-based learning, the sender can obtain a utility significantly larger than $\overline{\mathrm{OBJ}}^{\mathcal{R}}(\gamma)$ and $U^*$.*

The proof of this theorem is in Appendix E.4.

## 4 GENERALIZED PRINCIPAL-AGENT PROBLEMS WITH APPROXIMATE BEST RESPONSE

After presenting the reduction from learning to approximate best response, we now study generalized principal-agent problems with approximate best response. We will show that both the maxmin objectives $\underline{\mathrm{OBJ}}^{\mathcal{D}}(\delta)$, $\underline{\mathrm{OBJ}}^{\mathcal{R}}(\delta)$ and the maxmax objectives $\overline{\mathrm{OBJ}}^{\mathcal{D}}(\delta)$, $\overline{\mathrm{OBJ}}^{\mathcal{R}}(\delta)$ are close to the optimal principal objective $U^*$ in the best-response model when the degree $\delta$ of the agent's approximate best response is small, under some natural assumptions described below.

**Assumptions and notations.** We make some innocuous assumptions. First, the agent has no weakly dominated action:

**Assumption 1** (No Dominated Action). *An action $a_0 \in A$ of the agent is weakly dominated if there exists a mixed action $\alpha' \in \Delta(A \setminus \{a_0\})$ such that $v(x, \alpha') = \mathbb{E}_{a \sim \alpha'}[v(x,a)] \geq v(x, a_0)$ for all $x \in \mathcal{X}$. We assume that the agent has no weakly dominated action.*

**Claim 1.** *Assumption 1 implies: there exists a constant $G > 0$ such that, for any agent action $a \in A$, there exists a principal decision $x \in \mathcal{X}$ such that $v(x,a) - v(x, a') \geq G$ for every $a' \in A \setminus \{a\}$.*

The proof of this claim is in Appendix F.1. The constant $G > 0$ in Claim 1 is analogous to the concept of "inducibility gap" in Stackelberg games (Von Stengel & Zamir, 2004; Gan et al., 2023). In fact, Gan et al. (2023) show that, if the inducibility gap $G > \delta$, then the maximin approximate-best-response objective satisfies $\underline{\mathrm{OBJ}}^{\mathcal{D}}(\delta) \geq U^* - \frac{\delta}{G}$ in Stackelberg games. Our results will significantly generalize theirs to any generalized principal-agent problem, to randomized agent strategies, and to the maximax objectives $\overline{\mathrm{OBJ}}^{\mathcal{D}}(\delta)$, $\overline{\mathrm{OBJ}}^{\mathcal{R}}(\delta)$.

To present our results, we need to introduce a few more notions and assumptions. Let $\mathrm{diam}(\mathcal{X}; \|\cdot\|) = \max_{x_1, x_2 \in \mathcal{X}} \|x_1 - x_2\|$ be the diameter of the space $\mathcal{X}$, where $\|\cdot\|$ is some norm. For convenience we assume $\mathcal{X} \subseteq \mathbb{R}^d$ and use the $\ell_1$-norm $\|x\|_1 = \sum_{i=1}^d |x_{(i)}|$ or the $\ell_\infty$-norm $\|x\|_\infty = \max_{i=1}^d |x_{(i)}|$. For a generalized principal-agent problem with constraint $\sum_{s \in S} \pi_s x_s \in \mathcal{C}$, let $\partial \mathcal{X}$ be the boundary of $\mathcal{X}$ and let $\mathrm{dist}(\mathcal{C}, \partial \mathcal{X}) = \min_{c \in \mathcal{C}, x \in \partial X} \|c - x\|$ be the distance from $\mathcal{C}$ to the boundary of $\mathcal{X}$. We assume that $\mathcal{C}$ is away from the boundary of $\mathcal{X}$:

**Assumption 2** ($\mathcal{C}$ is in the interior of $\mathcal{X}$). $\mathrm{dist}(\mathcal{C}, \partial \mathcal{X}) > 0$.

**Assumption 3** (Bounded and Lipschitz utility). *The principal's utility function is bounded: $|u(x,a)| \leq B$, and $L$-Lipschitz in $x \in \mathcal{X}$: $|u(x_1, a) - u(x_2, a)| \leq L\|x_1 - x_2\|$.*

**Main results.** We now present the main results of this section: lower bounds on $\underline{\mathrm{OBJ}}^X(\delta)$ and upper bounds on $\overline{\mathrm{OBJ}}^X(\delta)$ in generalized principal-agent problems without and with constraints.

**Theorem 4.1** (Without constraint). *For an unconstrained generalized principal-agent problem, under Assumptions 1 and 3, for $0 \leq \delta < G$, we have*

- $\underline{\mathrm{OBJ}}^{\mathcal{D}}(\delta) \geq U^* - \mathrm{diam}(\mathcal{X})L\frac{\delta}{G}$.

- $\underline{\text{OBJ}}^{\mathcal{R}}(\delta) \geq U^* - 2\sqrt{\frac{2BL}{G}\text{diam}(\mathcal{X})\delta}$ *for* $\delta < \frac{\text{diam}(\mathcal{X})GL}{2B}$.

- $\overline{\text{OBJ}}^{\mathcal{D}}(\delta) \leq \overline{\text{OBJ}}^{\mathcal{R}}(\delta) \leq U^* + \text{diam}(\mathcal{X})L\frac{\delta}{G}$.

**Theorem 4.2** (With constraint). *For a generalized principal-agent problem with the constraint* $\sum_{s \in S} \pi_s x_s \in \mathcal{C}$, *under Assumptions 1, 2 and 3, for* $0 \leq \delta < \frac{G\text{dist}(\mathcal{C},\partial\mathcal{X})}{\text{diam}(\mathcal{X})}$, *we have*

- $\underline{\text{OBJ}}^{\mathcal{D}}(\delta) \geq U^* - \left(\text{diam}(\mathcal{X})L + 2B\frac{\text{diam}(\mathcal{X})}{\text{dist}(\mathcal{C},\partial\mathcal{X})}\right)\frac{\delta}{G}$.

- $\underline{\text{OBJ}}^{\mathcal{R}}(\delta) \geq U^* - 2\sqrt{\frac{2B}{G}\left(\text{diam}(\mathcal{X})L + 2B\frac{\text{diam}(\mathcal{X})}{\text{dist}(\mathcal{C},\partial\mathcal{X})}\right)\delta}$.

- $\overline{\text{OBJ}}^{\mathcal{D}}(\delta) \leq \overline{\text{OBJ}}^{\mathcal{R}}(\delta) \leq U^* + \left(\text{diam}(\mathcal{X})L + 2B\frac{\text{diam}(\mathcal{X})}{\text{dist}(\mathcal{C},\partial\mathcal{X})}\right)\frac{\delta}{G}$.

The expression "$\frac{\text{diam}(\mathcal{X})}{\text{dist}(\mathcal{C},\partial\mathcal{X})}\delta$" suggests that $\frac{1}{\text{dist}(\mathcal{C},\partial\mathcal{X})}$ is similar to a "condition number" (Renegar, 1994) that quantifies the "stability" of the principal-agent problem against the agent's approximate-best-responding behavior. When $\text{dist}(\mathcal{C},\partial\mathcal{X})$ is larger ($\mathcal{C}$ is further away from the boundary of $\mathcal{X}$), the condition number is smaller, the problem is more stable, and the $\delta$-best-response objectives $\underline{\text{OBJ}}^X(\delta)$ and $\overline{\text{OBJ}}^X(\delta)$ are closer to the best-response objective $U^*$.

**High-level idea: perturbation.** The high level idea to prove Theorems 4.1 and 4.2 is a perturbation argument. Consider proving the upper bounds on $\overline{\text{OBJ}}^{\mathcal{D}}(\delta)$ for example. Let $(\pi, \rho)$ be any pair of principal's strategy and agent's $\delta$-best-responding strategy. We perturb the principal's strategy $\pi$ slightly to be a strategy $\pi'$ to which $\rho$ is *exactly* best-responding (such a perturbation is possible due to Assumption 1). Since $\rho$ is best-responding to $\pi'$, the pair $(\pi', \rho)$ cannot give the principal a higher utility than $U^*$ (which is the optimal principal utility under the best-response model). This means that the original pair $(\pi, \rho)$ cannot give the principal a utility much higher than $U^*$, thus implying an upper bound on $\overline{\text{OBJ}}^{\mathcal{D}}(\delta)$. Extra care is needed when dealing with randomized strategies of the agent. See details in Appendix F.3.

**The bound** $\underline{\text{OBJ}}^{\mathcal{R}}(\delta) \geq U^* - O(\sqrt{\delta})$ **is tight.** We note that, in Theorems 4.1 and 4.2, the maxmin objective with randomized agent strategies is bounded by $\underline{\text{OBJ}}^{\mathcal{R}}(\delta) \geq U^* - O(\sqrt{\delta})$, while the objective with deterministic agent strategies is bounded by $\underline{\text{OBJ}}^{\mathcal{D}}(\delta) \geq U^* - O(\delta)$. This is *not* because our analysis is not tight. In fact, the squared root bound $U^* - \Theta(\sqrt{\delta})$ for $\underline{\text{OBJ}}^{\mathcal{R}}(\delta)$ is tight. We prove this by giving an example where $\underline{\text{OBJ}}^{\mathcal{R}}(\delta) \leq U^* - \Omega(\sqrt{\delta})$. Consider the following classical Bayesian persuasion example:

**Example 4.1.** *There are 2 states* $\Omega = \{\text{Good}, \text{Bad}\}$, *2 actions* $A = \{a, b\}$, *with utility matrices*

| sender | a | b |
|--------|---|---|
| Good | 1 | 0 |
| Bad | 1 | 0 |

| receiver | a | b |
|----------|---|---|
| Good | 1 | 0 |
| Bad | $-1$ | 0 |

*The prior probability of* Good *state is* $\mu_0 < \frac{1}{2}$, *so the receiver takes action b by default. In this example, for* $\delta < \frac{\mu_0}{2}$, $\underline{\text{OBJ}}^{\mathcal{R}}(\delta) \leq U^* - 2\sqrt{2\mu_0\delta} + \delta = U^* - \Omega(\sqrt{\delta})$. *See Appendix F.2 for a proof.*

## 5 APPLICATIONS TO SPECIFIC PRINCIPAL-AGENT PROBLEMS

We apply the general results in Section 3 and 4 to derive concrete results for three specific principal-agent problems: Bayesian persuasion, Stackelberg games, and contract design.

**Bayesian persuasion.** As noted in Section 2, Bayesian persuasion is a generalized principal-agent problem with constraint $\sum_{s \in S} \pi_s x_s \in \mathcal{C} = \{\mu_0\}$ where each $x_s = \mu_s = (\mu_s(\omega))_{\omega \in \Omega} \in \mathcal{X} = \Delta(\Omega)$ is a posterior belief. Suppose the principal's utility is bounded: $|u(\omega, a)| \leq B$. Then, the principal's utility function $u(\mu_s, a) = \sum_{\omega \in \Omega} \mu_s(\omega)u(\omega, a)$ is $(L = B)$-Lipschitz in $\mu_s$ (under $\ell_1$-norm), so Assumption 3 is satisfied. Suppose the prior $\mu_0$ has positive probability for every $\omega \in \Omega$, and let $p_0 = \min_{\omega \in \Omega} \mu_0(\omega) > 0$. Then, we have the distance

$$\text{dist}(\mathcal{C}, \partial X) = \min\left\{\|\mu_0 - \mu\|_1 : \mu \in \Delta(\Omega) \text{ s.t. } \mu(\omega) = 0 \text{ for some } \omega \in \Omega\right\} \geq p_0 > 0,$$

so Assumption 2 is satisfied. The diameter satisfies $\mathrm{diam}(\mathcal{X}; \ell_1) = \max_{\mu_1, \mu_2 \in \Delta(\Omega)} \|\mu_1 - \mu_2\|_1 \leq 2$. Finally, we assume Assumption 1 (no dominated action for the agent). Then, Theorem 4.2 gives bounds on the approximate-best-response objectives in Bayesian persuasion:

**Corollary 5.1** (Bayesian persuasion with approximate best response). *For $0 \leq \delta < \frac{Gp_0}{2}$,*

- $\underline{\mathrm{OBJ}}^{\mathcal{D}}(\delta) \geq U^* - 2B(1 + \frac{2}{p_0})\frac{\delta}{G}$, *and* $\underline{\mathrm{OBJ}}^{\mathcal{R}}(\delta) \geq U^* - 4B\sqrt{(1 + \frac{2}{p_0})\frac{\delta}{G}}$.

- $\overline{\mathrm{OBJ}}^{\mathcal{D}}(\delta) \leq \overline{\mathrm{OBJ}}^{\mathcal{R}}(\delta) \leq U^* + 2B(1 + \frac{2}{p_0})\frac{\delta}{G}$.

Further applying Theorem 3.1 and 3.4, we obtain the central result for our motivating problem, persuasion with a learning agent:

**Corollary 5.2** (Persuasion with a learning agent). *Suppose $T$ is sufficiently large such that $\frac{\mathrm{CReg}(T)}{T} < \frac{Gp_0}{2}$ and $\frac{\mathrm{CSReg}(T)}{T} < \frac{Gp_0}{2}$, then*

- *with a contextual no-regret learning agent, the principal can obtain utility at least*

$$\frac{1}{T}\mathbb{E}\Big[\sum_{t=1}^{T} u(x^t, a^t)\Big] \geq \underline{\mathrm{OBJ}}^{\mathcal{R}}\big(\tfrac{\mathrm{CReg}(T)}{T}\big) \geq U^* - 4B\sqrt{(1 + \tfrac{2}{p_0})\tfrac{1}{G}}\sqrt{\tfrac{\mathrm{CReg}(T)}{T}} \qquad (8)$$

  *using a fixed signaling schemes in all rounds.*

- *with a contextual no-swap-regret learning agent, the principal's obtainable utility is at most*

$$\frac{1}{T}\mathbb{E}\Big[\sum_{t=1}^{T} u(x^t, a^t)\Big] \leq \overline{\mathrm{OBJ}}^{\mathcal{D}}\big(\tfrac{\mathrm{CSReg}(T)}{T}\big) \leq U^* + 2B(1 + \tfrac{2}{p_0})\tfrac{1}{G}\tfrac{\mathrm{CSReg}(T)}{T} \qquad (9)$$

  *even using adaptive signaling schemes.*

Result (9) is particularly interesting. First, it shows that the principal cannot exploit a no-swap-regret learning agent beyond $U^* + o(1)$ even if the principal has informational advantage (knowing the state $\omega$). Second, this result still holds even if we allow the principal to see the agent's strategy $\rho^t$ before choosing the signaling scheme $\pi^t$. The principal still cannot exploit the agent in this case.

**Stackelberg games and contract design.** When applied to Stackelberg games and contract design, our results show that the principal can obtain $U^* - O(\sqrt{\frac{\mathrm{CReg}(T)}{T}})$ against a no-regret agent and no more than $U^* + O(\frac{\mathrm{CSReg}(T)}{T})$ against a no-swap-regret agent. These results refine the $U^* + o(1)$ and $U^* - o(1)$ bounds in Deng et al. (2019); Guruganesh et al. (2024). See Appendix D for details. This demonstrates the generality and usefulness of our framework.

## 6 DISCUSSION

In summary, our work provides an explicit characterization of the principal's achievable utility in generalized principal-agent problems with a contextual no-swap-regret learning agent. It is an asymmetric range $\big[U^* - O(\sqrt{\frac{\mathrm{CSReg}(T)}{T}}), U^* + O(\frac{\mathrm{CSReg}(T)}{T})\big]$. We show that this conclusion holds in all generalized principal-agent problems where the agent does not have private information, in particular including Bayesian persuasion where the principal is privately informed. As we mentioned in the Introduction, the upper bound $U^* + O(\frac{\mathrm{CSReg}(T)}{T})$ does not hold when the agent has private information or does certain types of no-regret but not no-swap-regret learning. Deriving the exact upper bound in the latter cases is an interesting direction for future work.

Other directions for future work include, for example, relaxing the assumption that the principal has perfect knowledge of the environment – what if both principal and agent are learning players? And what if the environment is non-stationary, like a Markovian environment (Jain & Perchet, 2024) or an adversarial dynamic environment (Camara et al., 2020)? In unknown or non-stationary environments, the benchmark $U^*$ needs to be redefined, and a joint design of both players' learning algorithms might be interesting.

ACKNOWLEDGMENTS

Yiling is supported by the National Science Foundation under grant no. IIS-2147187 and by Amazon. Tao is supported by a Siebel PhD scholarship.

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

## A    ADDITIONAL RELATED WORKS

Learning agents have been studied in principal-agent problems like auctions (Braverman et al., 2018; Cai et al., 2024; Rubinstein & Zhao, 2024; Kumar et al., 2024), bimatrix Stackelberg games (Deng et al., 2019; Mansour et al., 2022; Arunachaleswaran et al., 2024), contract design (Guruganesh et al., 2024; Scheid et al., 2024), and Bayesian persuasion (Lin et al., 2023; Jain & Perchet, 2024). These problems belong to the class of *generalized principal-agent problems* (Myerson, 1982; Gan et al., 2024). We thus propose a general framework of generalized principal-agent problem with a learning agent, which encompasses several previous models, refines previous results, and provides new results.

Camara et al. (2020) also propose a general framework of principal-agent problems with learning players, but has two key differences with ours: (1) They drop the common prior assumption while we still keep it. This assumption allows us to compare the principal's utility in the learning model with the classic model with common prior. (2) Their principal has commitment power, which is reasonable in, e.g., auction design, but less realistic in information design where the principal's strategy is a signaling scheme. Our principal does not commit.

Deng et al. (2019) show that the follower's no-swap-regret learning can cap the leader's utility at $U^* + o(1)$ in Stackelberg games. We find that this conclusion holds for all generalized principal-agent problems where the agent does not have private information. This conclusion does not hold when the agent is privately informed, as shown by Mansour et al. (2022) in Bayesian Stackelberg games. We view our work as characterizing the largest class of games under which this conclusion holds.

The literature on information design (Bayesian persuasion) has investigated various relaxations of the strong rationality assumptions in the classic models. For the sender, known prior (Camara et al., 2020; Ziegler, 2020; Zu et al., 2021; Kosterina, 2022; Wu et al., 2022; Dworczak & Pavan, 2022; Harris et al., 2023; Lin & Li, 2025) and known utility (Babichenko et al., 2021; Castiglioni et al., 2020; Feng et al., 2022; Bacchiocchi et al., 2024) are relaxed. For the receiver, the receiver may make mistakes in Bayesian updates (de Clippel & Zhang, 2022), be risk-conscious (Anunrojwong et al., 2023), do quantal response (Feng et al., 2024) or approximate best response (Yang & Zhang, 2024). Independently and concurrently of us, Jain & Perchet (2024) also study Bayesian persuasion with a learning agent. Their work has a few differences with us: First, their model is a general Bayesian persuasion model with imperfect and non-stationary dynamics for the state of the world. Our model generalizes Bayesian persuasion in another direction (namely, generalized principal-agent problems), while still assuming a perfect and stationary environment. Second, their results are qualitatively similar to our Result 1 and Result 4, while our results are more quantitative and precise. Third, we additionally show that no-swap-regret learning can cap the sender's utility (Result 2).

As our problem reduces to generalized principal-agent problems with approximate best response, our work is also related to recent works on approximately-best-responding agents in Stackelberg games (Gan et al., 2023) and Bayesian persuasion (Yang & Zhang, 2024). We focus on the range of payoff that can be obtained by a computationally-unbounded principal, ignoring the computational aspect considered by Gan et al. (2023); Yang & Zhang (2024). Besides the "maxmin/robust" objective, we also study the "maxmax" objective where the agent approximately best responds *in favor of* the principal, which is usually not studied in the literature.

## B    DETAILS ABOUT CONTEXTUAL NO-(SWAP-)REGRET ALGORITHMS

Contextual no-(swap-)regret algorithms can be constructed by running an ordinary no-(swap-)regret algorithm for each context independently. Since algorithms with $O(\sqrt{T})$ (swap-)regret exist under bandit feedback (Audibert & Bubeck, 2010; Ito, 2020), they lead to algorithms with $O(\sqrt{|S|T})$ contextual (swap-)regret. This is formalized by the following proposition:

**Proposition B.1.** *There exist learning algorithms with contextual regret* $\mathrm{CReg}(T) = O(\sqrt{|A||S|T})$ *and contextual swap-regret* $\mathrm{CSReg}(T) = O(|A|\sqrt{|S|T})$. *They can be constructed by running an ordinary no-(swap-)regret multi-armed bandit algorithm for each context independently.*

We prove Proposition B.1 in the rest of this section.

Let $\mathcal{A}$ be an arbitrary no-regret (no-swap-regret) learning algorithm for a multi-armed bandit (MAB) problem with $|A|$ arms. There exist such algorithms with regret $O(\sqrt{T|A|\log|A|})$ (variants of Exp3 (Auer et al., 2002)) and even $O(\sqrt{T|A|})$ (doubling trick + polyINF (Audibert & Bubeck, 2010)) for any time horizon $T > 0$. By swap-to-external regret reductions, they can be converted to multi-armed bandit algorithms with swap regret $O(\sqrt{T|A|^3 \log|A|})$ (Blum & Mansour, 2007) and $O(|A|\sqrt{T})$ (Ito, 2020). We then convert $\mathcal{A}$ into a contextual no-regret (contextual no-swap-regret) algorithm, in the following way:

---

**Algorithm 1:** Convert any MAB algorithm to a contextual MAB algorithm

---

**Input:** MAB algortihm $\mathcal{A}$. Arm set $A$. Context set $S$.
Instantiate $|S|$ copies $\mathcal{A}_1, \ldots, \mathcal{A}_{|S|}$ of $\mathcal{A}$, and initialize their round number by
$\quad t_1 = \cdots = t_{|S|} = 0$.
**for** *round $t = 1, 2, \ldots$* **do**
$\quad$ Receive context $s^t$. Call $\mathcal{A}_{s^t}$ to obtain an action $a^t$.
$\quad$ Play $a^t$ and obtain feedback (which includes the reward $v^t(a^t)$ of action $a^t$).
$\quad$ Feed the feedback to $\mathcal{A}_{s^t}$. Increase its round number $t_{s^t}$ by 1.
**end**

---

**Proposition B.2.** *The contextual regret of Algorithm 1 is at most*

$$\mathrm{CReg}(T) \leq \max\Big\{ \sum_{s=1}^{|S|} \mathrm{Reg}(T_s) \,\Big|\, T_1 + \cdots + T_{|S|} = T \Big\},$$

*where $\mathrm{Reg}(T_s)$ is the regret of $\mathcal{A}$ for time horizon $T_s$.*

*The contextual swap-regret of Algorithm 1 is at most*

$$\mathrm{CSReg}(T) \leq \max\Big\{ \sum_{s=1}^{|S|} \mathrm{SReg}(T_s) \,\Big|\, T_1 + \cdots + T_{|S|} = T \Big\},$$

*where $\mathrm{SReg}(T_s)$ is the swap-regret of $\mathcal{A}$ for time horizon $T_s$.*

*When plugging in $\mathrm{Reg}(T_s) = O(\sqrt{|A|T_s})$, we obtain $\mathrm{CReg}(T) \leq O(\sqrt{|A||S|T})$.*

*When plugging in $\mathrm{SReg}(T_s) = O(|A|\sqrt{T_s})$, we obtain $\mathrm{CSReg}(T) \leq O(|A|\sqrt{|S|T})$.*

*Proof.* The contextual regret of Algorithm 1 is

$$\mathrm{CReg}(T) = \max_{d:S \to A} \mathbb{E}\Big[ \sum_{t=1}^{T} \big(v^t(d(s^t)) - v^t(a^t)\big) \Big]$$

$$= \max_{d:S \to A} \mathbb{E}\Big[ \sum_{s=1}^{|S|} \sum_{t:s^t=s} \big(v^t(d(s)) - v^t(a^t)\big) \Big]$$

$$\leq \sum_{s=1}^{|S|} \max_{a' \in A} \mathbb{E}\Big[ \sum_{t:s^t=s} \big(v^t(a') - v^t(a^t)\big) \Big]$$

$$\leq \sum_{s=1}^{|S|} \mathbb{E}_{T_s}\big[\mathrm{Reg}(T_s)\big] \quad \text{where } T_s \text{ is the number of rounds where } s^t = s$$

$$\leq \max\Big\{ \sum_{s=1}^{|S|} \mathrm{Reg}(T_s) \,\Big|\, T_1 + \cdots + T_{|S|} = T \Big\}.$$

When $\mathrm{Reg}(T_s) = O(\sqrt{|A|T_s})$, by Jensen's inequality we obtain

$$\mathrm{CReg}(T) \leq \sum_{s=1}^{|S|} O(\sqrt{|A|T_s}) \leq O(\sqrt{|A|})\sqrt{|S|}\sqrt{\sum_{s=1}^{|S|} T_s} = O(\sqrt{|A||S|T}).$$

The argument for contextual swap-regret is similar:

$$\begin{aligned}
\text{CSReg}(T) &= \max_{d:S \times A \to A} \mathbb{E}\Big[\sum_{t=1}^{T}\big(v^t(d(s^t, a^t)) - v^t(a^t)\big)\Big] \\
&= \max_{d:S \times A \to A} \mathbb{E}\Big[\sum_{s=1}^{|S|}\sum_{t:s^t=s}\big(v^t(d(s, a^t)) - v^t(a^t)\big)\Big] \\
&\le \sum_{s=1}^{|S|} \max_{d':A \to A} \mathbb{E}\Big[\sum_{t:s^t=s}\big(v^t(d'(a^t)) - v^t(a^t)\big)\Big] \\
&\le \sum_{s=1}^{|S|} \mathbb{E}_{T_s}\big[\text{SReg}(T_s)\big] \quad \text{where } T_s \text{ is the number of rounds where } s^t = s \\
&\le \max\Big\{\sum_{s=1}^{|S|} \text{SReg}(T_s) \,\Big|\, T_1 + \cdots + T_{|S|} = T\Big\}.
\end{aligned}$$

When $\text{SReg}(T_s) = O(|A|\sqrt{T_s})$, by Jensen's inequality we obtain

$$\text{CSReg}(T) \le \sum_{s=1}^{|S|} O(|A|\sqrt{T_s}) \le O(|A|)\sqrt{|S|}\sqrt{\sum_{s=1}^{|S|} T_s} = O(|A|\sqrt{|S|T}).$$

$\square$

## C  EXAMPLE OF APPROXIMATELY BEST RESPONDING AGENTS

Our model of approximately-best-responding agent (Section 3.1) includes, for example, two other models in the Bayesian persuasion literature that also relax the agent's Bayesian rationality assumption: the quantal response model (proposed by (McKelvey & Palfrey, 1995) in normal-form games and studied by (Feng et al., 2024) in Bayesian persuasion) and a model where the agent makes mistakes in Bayesian update (de Clippel & Zhang, 2022):

**Example C.1.** *Assume that the receiver's utility is in* $[0, 1]$*. In Bayesian persuasion, the following receiver strategies are* $\delta$*-best-responding:*

- Quantal response: *given signal* $s \in S$*, the agent chooses action* $a \in A$ *with probability* $\frac{\exp(\lambda v(\mu_s, a))}{\sum_{a' \in A} \exp(\lambda v(\mu_s, a'))}$*, with* $\lambda > 0$*. This strategy belongs to* $\mathcal{R}_\delta(\pi)$ *with* $\delta = \frac{1 + \log(|A|\lambda)}{\lambda}$*.*

- Inaccurate belief: *given signal* $s \in S$*, the agent forms some posterior* $\mu'_s$ *that is different yet close to the true posterior* $\mu_s$ *in total variation distance* $d_{\text{TV}}(\mu'_s, \mu_s) \le \varepsilon$*. The agent picks an optimal action for* $\mu'_s$*. This strategy belongs to* $\mathcal{D}_{2\varepsilon}(\pi)$*.*

*Proof.* Consider the quantal response model. Let $\gamma = \frac{\log(|A|\lambda)}{\lambda}$. Given signal $s$, with posterior $\mu_s$, we say an action $a \in A$ is *not* $\gamma$-optimal for posterior $\mu_s$ if

$$v(\mu_s, a_s^*) - v(\mu_s, a) \ge \gamma$$

where $a_s^*$ is an optimal action for $\mu_s$. The probability that the receiver chooses not $\gamma$-optimal action $a$ is at most:

$$\frac{\exp(\lambda v(\mu_s, a))}{\sum_{a \in A} \exp(\lambda v(\mu_s, a))} \le \frac{\exp(\lambda v(\mu_s, a))}{\exp(\lambda v(\mu_s, a_s^*))} = \exp\Big(-\lambda\big[v(\mu_s, a_s^*) - v(\mu_s, a)\big]\Big) \le \exp(-\lambda\gamma) = \frac{1}{|A|\lambda}.$$

By a union bound, the probability that the receiver chooses any not $\gamma$-approxiamtely optimal action is at most $\frac{1}{\lambda}$. So, the expected loss of utility of the receiver due to not taking the optimal action is at most

$$\Big(1 - \frac{1}{\lambda}\Big) \cdot \gamma + \frac{1}{\lambda} \cdot 1 \le \frac{\log(|A|\lambda) + 1}{\lambda}$$

This means that the quantal response strategy is a $\frac{\log(|A|\lambda)+1}{\lambda}$-best-responding randomized strategy.

Consider inaccurate belief. Given signal $s$, the receiver has belief $\mu'_s$ with total variation distance $d_{\mathrm{TV}}(\mu'_s, \mu_s) \leq \varepsilon$ to the true posterior $\mu_s$. For any action $a \in A$, the difference of expected utility of action $a$ under beliefs $\mu'_s$ and $\mu_s$ is at most $\varepsilon$:

$$\left| \mathbb{E}_{\omega \sim \mu'_s}[v(\omega, a)] - \mathbb{E}_{\omega \sim \mu_s}[v(\omega, a)] \right| \leq d_{\mathrm{TV}}(\mu'_s, \mu_s) \leq \varepsilon.$$

So, the optimal action for $\mu'_s$ is a $2\varepsilon$-optimal action for $\mu_s$. This means that the receiver strategy is a deterministic $2\varepsilon$-best-responding strategy. $\qquad\square$

# D  ADDITIONAL DETAILS ON APPLICATIONS TO SPECIFIC PRINCIPAL-AGENT PROBLEMS

## D.1  STACKELBERG GAMES

In a Stackelberg game, the principal (leader), having a finite action set $B$, first commits to a mixed strategy $x = (x_{(b)})_{b \in B} \in \Delta(B)$, which is a distribution over actions. So the principal's decision space $\mathcal{X}$ is $\Delta(B)$. The agent (follower) then takes an action $a \in A$ in response to $x$. The (expected) utilities for the two players are $u(x, a) = \sum_{b \in B} x_{(b)} u(b, a)$ and $v(x, a) = \sum_{b \in B} x_{(b)} u(b, a)$. The signal $s$ can (but not necessarily) be an action that the principal recommends the agent to take.

Assume bounded utility $|u(b, a)| \leq B$. Then, the principal's utility function $u(x, a)$ is bounded in $[-B, B]$ and $(L = B)$-Lipschitz in $x$. The diameter $\mathrm{diam}(\mathcal{X}) = \max_{x_1, x_2 \in \Delta(B)} \|x_1 - x_2\|_1 \leq 2$. Applying the theorem for unconstrained generalized principal-agent problems (Theorem 4.1) and the theorems for learning agent (Theorem 3.1 and 3.4), we obtain:

**Corollary D.1** (Stackelberg game with a learning agent). *Suppose $T$ is sufficiently large such that $\frac{\mathrm{CReg}(T)}{T} < G$ and $\frac{\mathrm{CSReg}(T)}{T} < G$, then:*

- *with a contextual no-regret learning agent, the principal can obtain utility $\frac{1}{T}\mathbb{E}\big[\sum_{t=1}^{T} u(x^t, a^t)\big] \geq \underline{\mathrm{OBJ}}^{\mathcal{R}}\big(\frac{\mathrm{CReg}(T)}{T}\big) \geq U^* - \frac{4B}{\sqrt{G}}\sqrt{\frac{\mathrm{CReg}(T)}{T}}$ using a fixed strategy in all rounds.*

- *with a contextual no-swap-regret learning agent, the principal cannot obtain utility more than $\frac{1}{T}\mathbb{E}\big[\sum_{t=1}^{T} u(x^t, a^t)\big] \leq \overline{\mathrm{OBJ}}^{\mathcal{D}}\big(\frac{\mathrm{CSReg}(T)}{T}\big) \leq U^* + \frac{2B}{G}\frac{\mathrm{CSReg}(T)}{T}$ even using adaptive strategies.*

The conclusion that the principal can obtain utility at least $U^* - o(1)$ against a no-regret learning agent and no more than $U^* + o(1)$ against a no-swap-regret agent in Stackelberg games was proved by (Deng et al., 2019). Our Corollary D.1 reproduces this conclusion and moreover provides bounds on the $o(1)$ terms, namely, $U^* - O(\sqrt{\frac{\mathrm{CReg}(T)}{T}})$ and $U^* + O(\frac{\mathrm{CSReg}(T)}{T})$. This demonstrates the generality and usefulness of our framework.

## D.2  CONTRACT DESIGN

In contract design, there is a finite outcome space $O = \{r_1, \ldots, r_d\}$ where each $r_i \in \mathbb{R}$ is a monetary reward to the principal. When the agent takes action $a \in A$, outcome $r_i$ will happen with probability $p_{ai} \geq 0$, $\sum_{i=1}^{d} p_{ai} = 1$. The principal cannot observe the action taken by the agent but can observe the realized outcome. The principal's decision space $\mathcal{X}$ is the set of contracts, where a contract $x = (x_{(i)})_{i=1}^{d} \in [0, +\infty]^d$ is a vector that specifies the payment to the agent for each possible outcome. So, if the agent takes action $a$ under contract $x$, the principal obtains expected utility

$$u(x, a) = \sum_{i=1}^{d} p_{ai}(r_i - x_{(i)})$$

and the agent obtains $v(x, a) = \sum_{i=1}^{d} p_{ai} x_{(i)} - c_a$, where $c_a \geq 0$ is the cost of action $a \in A$ for the agent. The signal $s$ can (but not necessarily) be an action that the principal recommends the agent to take. The principal's decision space $\mathcal{X} \subseteq [0, +\infty]^d$ in contract design, however, may be unbounded

and violate the requirement of bounded diameter $\text{diam}(\mathcal{X})$ that we need. We have two remedies for this.

The first remedy is to require the principal's payment to the agent be upper bounded by some constant $P < +\infty$, so $0 \leq x_{(i)} \leq P$ and $\mathcal{X} = [0, P]^d$. Under this requirement and the assumption of bounded reward $|r_i| \leq R$, the principal's utility becomes bounded by $|u(x, a)| \leq \sum_{i=1}^d p_{ai}(R + P) = R + P = B$ and $(L = 1)$-Lipschitz under $\ell_\infty$-norm:

$$|u(x_1, a) - u(x_2, a)| = \Big| \sum_{i=1}^d p_{ai}(x_{1(i)} - x_{2(i)}) \Big| \leq \max_{i=1}^d |x_{1(i)} - x_{2(i)}| \sum_{i=1}^d p_{ai} = \|x_1 - x_2\|_\infty. \quad (10)$$

And the diameter of $\mathcal{X}$ is bounded by (under $\ell_\infty$-norm)

$$\text{diam}(\mathcal{X}; \ell_\infty) = \max_{x_1, x_2 \in \mathcal{X}} \|x_1 - x_2\|_\infty = \max_{x_1, x_2 \in [0,P]^d} \max_{i=1}^d |x_{1(i)} - x_{2(i)}| \leq P. \quad (11)$$

Now, we can apply the theorem for unconstrained generalized principal-agent problems (Theorem 4.1) and the theorems for learning agent (Theorem 3.1 and Theorem 3.4) to obtain:

**Corollary D.2** (Contract design (with bounded payment) with a learning agent)**.** *Suppose $T$ is sufficiently large such that $\frac{\text{CReg}(T)}{T} < \frac{PG}{2(R+P)}$ and $\frac{\text{CSReg}(T)}{T} < G$, then:*

- *with a contextual no-regret learning agent, the principal can obtain utility at least $\frac{1}{T}\mathbb{E}\big[ \sum_{t=1}^T u(x^t, a^t) \big] \geq \underline{\text{OBJ}}^{\mathcal{R}}\big( \frac{\text{CReg}(T)}{T} \big) \geq U^* - 2\sqrt{\frac{2(R+P)P}{G}}\sqrt{\frac{\text{CReg}(T)}{T}}$ using a fixed contract in all rounds.*

- *with contextual a no-swap-regret learning agent, the principal cannot obtain utility more than $\frac{1}{T}\mathbb{E}\big[ \sum_{t=1}^T u(x^t, a^t) \big] \leq \overline{\text{OBJ}}^{\mathcal{D}}\big( \frac{\text{CSReg}(T)}{T} \big) \leq U^* + \frac{P}{G}\frac{\text{CSReg}(T)}{T}$ even using adaptive contracts.*

The second remedy is to write contract design as a generalized principal-agent problem in another way. Let $\tilde{x} = (\tilde{x}_{(a)})_{a \in A} \in [0, +\infty]^{|A|}$ be a vector recording the *expected payment* from the principal to the agent for each action $a \in A$:

$$\tilde{x}_{(a)} = \sum_{i=1}^d p_{ai} x_{(i)}. \quad (12)$$

And let $\tilde{r}_{(a)}$ be the expected reward of action $a$, $\tilde{r}_{(a)} = \sum_{i=1}^d p_{ai} r_i$. Then, the principal and the agent's utility can be rewritten as functions of $\tilde{x}$ and $a$:

$$u(\tilde{x}, a) = \tilde{r}_{(a)} - \tilde{x}_{(a)}, \qquad v(\tilde{x}, a) = \tilde{x}_{(a)} - c_a, \quad (13)$$

which are linear (strictly speaking, affine) in $\tilde{x} \in \tilde{\mathcal{X}}$. Assuming bounded reward $|\tilde{r}_{(a)}| \leq R$, we can without loss of generality assume that the expected payment $\tilde{x}_{(a)}$ is bounded by $R$ as well, because otherwise the principal will get negative utility. So, the principal's decision space can be restricted to

$$\tilde{\mathcal{X}} = \Big\{ \tilde{x} \mid \exists \, x \in [0, +\infty]^d \text{ such that } \tilde{x}_{(a)} = \sum_{i=1}^d p_{ai} x_{(i)} \text{ for every } a \in A \Big\} \cap [0, R]^{|A|}, \quad (14)$$

which is convex and has bounded diameter (under $\ell_\infty$ norm)

$$\text{diam}(\tilde{\mathcal{X}}; \ell_\infty) \leq \text{diam}([0, R]^{|A|}; \ell_\infty) = R. \quad (15)$$

The utility function $u(\tilde{x}, a)$ is bounded by $2R$ and $(L = 1)$-Lipschitz (under $\ell_\infty$ norm):

$$|u(\tilde{x}_1, a) - u(\tilde{x}_2, a)| = |\tilde{x}_{1(a)} - \tilde{x}_{2(a)}| \leq \max_{a \in A} |\tilde{x}_{1(a)} - \tilde{x}_{2(a)}| = \|\tilde{x}_1 - \tilde{x}_2\|_\infty. \quad (16)$$

Thus, we can apply the theorem for unconstrained generalized principal-agent problems (Theorem 4.1) and the theorems for learning agent (Theorem 3.1 and Theorem 3.4) to obtain:

**Corollary D.3** (Contract design with a learning agent)**.** *Suppose $T$ is sufficiently large such that $\frac{\text{CReg}(T)}{T} < \frac{G}{2}$ and $\frac{\text{CSReg}(T)}{T} < G$, then:*

- *with a contextual no-regret learning agent, the principal can obtain utility at least* $\frac{1}{T}\mathbb{E}\big[\sum_{t=1}^{T}u(x^t,a^t)\big] \geq \underline{\text{OBJ}}^{\mathcal{R}}\big(\frac{\text{CReg}(T)}{T}\big) \geq U^* - \frac{4R}{\sqrt{G}}\sqrt{\frac{\text{CReg}(T)}{T}}$ *using a fixed contract in all rounds.*

- *with a contextual no-swap-regret learning agent, the principal cannot obtain utility more than* $\frac{1}{T}\mathbb{E}\big[\sum_{t=1}^{T}u(x^t,a^t)\big] \leq \overline{\text{OBJ}}^{\mathcal{D}}\big(\frac{\text{CSReg}(T)}{T}\big) \leq U^* + \frac{R}{G}\frac{\text{CSReg}(T)}{T}$ *even using adaptive contracts.*

Providing the quantitative lower and upper bounds, the above results refine the result in (Guruganesh et al., 2024) that the principal can obtain utility at least $U^* - o(1)$ against a no-regret learning agent and no more than $U^* + o(1)$ against a no-swap-regret agent. This again demonstrates the versatility of our general framework.

## E    MISSING PROOFS FROM SECTION 3

### E.1    PROOF OF LEMMA 3.2

Since $\pi^t = \pi$ is fixed, we have $\pi_s^t = \pi_s$ and $x_s^t = x_s$, $\forall s \in S$. The regret of the receiver not deviating according to $d$ is:

$$\frac{1}{T}\mathbb{E}\Big[\sum_{t=1}^{T}\Big(v(x^t,d(s^t)) - v(x^t,a^t)\Big)\Big] = \frac{1}{T}\sum_{t=1}^{T}\sum_{s\in S}\pi_s^t\sum_{a\in A}p_{a|s}^t\Big(v(x_s^t,d(s)) - v(x_s^t,a)\Big)$$

$$= \sum_{s\in S}\pi_s\sum_{a\in A}\frac{\sum_{t=1}^{T}p_{a|s}^t}{T}\Big(v(x_s,d(s)) - v(x_s,a)\Big)$$

$$= \sum_{s\in S}\pi_s v(x_s,d(s)) \; - \; \sum_{s\in S}\pi_s\sum_{a\in A}\rho(a|s)v(x_s,a) \; = \; V(\pi,d) - V(\pi,\rho).$$

Here, $d$ is interpreted as an agent strategy that deterministically takes action $d(s)$ for signal $s$.

By a similar derivation, we see that the principal's expected utility is equal to $\frac{1}{T}\mathbb{E}\Big[\sum_{t=1}^{T}u(x^t,a^t)\Big] = \sum_{s\in S}\pi_s\sum_{a\in A}\frac{\sum_{t=1}^{T}p_{a|s}^t}{T}u(x_s,a) = U(\pi,\rho)$, which proves the lemma.

### E.2    PROOF OF THEOREM 3.3

We prove this theorem for no-swap-regret learning algorithms. The argument for no-regret learning algorithms is analogous.

Fix the principal's adaptive strategy $\sigma = (\sigma^t)_{t=1}^{T}$ for the $T$ rounds, where each $\sigma^t$ is a mapping from the history $h^{t-1} = (s^i,a^i)_{i=1}^{t-1}$ (including past signals and actions) to a single-round strategy $\pi^t$ for round $t$. Given any function $\text{CSReg}(T)$, let $\delta = \frac{\text{CSReg}(T)}{2T}$. We construct the following algorithm $\mathcal{A}$ for the agent: at each round $t$, given history $h^{t-1} = (s^i,a^i)_{i=1}^{t-1}$,

- If the single-round strategy chosen by the principal at round $t$ is equal to $\pi^t = \sigma^t(h^{t-1})$, then the agent plays a strategy $\rho^t \in \arg\min_{\rho\in\mathcal{R}_\delta(\pi^t)}U(\pi^t,\rho)$, namely, a randomized $\delta$-best-responding strategy that minimizes the principal's utility.

- If the single-round strategy chosen by the principal at round $t$ is not equal to $\pi^t = \sigma^t(h^{t-1})$, then the agent switches to any existing contextual no-swap-regret algorithm with swap regret at most $\frac{\text{CSReg}(T)}{2}$ (see Proposition B.1 for examples of such algorithms).

We show that the agent's algorithm has swap regret at most $\text{CSReg}(T)$ no matter what strategy the principal uses:

- If the principal keeps using strategy $\sigma$, namely, at each round $t$ the principal uses single-round strategy $\pi^t = \sigma^t(h^{t-1})$, denoted by $\pi^t = \{(\pi_s^t, x_s^t)\}_{s\in S}$, then the agent will respond by strategy $\rho^t$. For any deviation function $d : S \times A \to A$, the expected regret of the agent not deviating according

to $d$ in this round is

$$
\begin{aligned}
\mathbb{E}[v(x^t, d(s^t, a^t)) - v(x^t, a^t)] &= \mathbb{E}_{h^{t-1}}\Big[\sum_{s \in S} \pi_s^t \sum_{a \in A} \rho^t(a|s)\Big(v(x_s^t, d(s, a)) - v(x_s^t, a)\Big)\Big] \\
&= \mathbb{E}_{h^{t-1}}\Big[\sum_{s \in S} \pi_s^t \sum_{a \in A} \rho^t(a|s)v(x_s^t, d(s, a)) - \sum_{s \in S} \pi_s^t \sum_{a \in A} \rho^t(a|s)v(x_s^t, a)\Big] \\
&\leq \mathbb{E}_{h^{t-1}}\Big[\sum_{s \in S} \pi_s^t \max_{a \in A} v(x_s^t, a) - \sum_{s \in S} \pi_s^t \sum_{a \in A} \rho^t(a|s)v(x_s^t, a)\Big] \\
&= \mathbb{E}_{h^{t-1}}\Big[\sum_{s \in S} \pi_s^t \max_{a \in A} v(x_s^t, a) - V(\pi^t, \rho^t)\Big] \\
&\leq \mathbb{E}_{h^{t-1}}\Big[\delta\Big] \quad \text{because by definition, } \rho^t \in \mathcal{R}_\delta(\pi^t) \iff V(\pi^t, \rho^t) \geq \sum_{s \in S} \pi_s^t \max_{a \in A} v(x_s^t, a) - \delta.
\end{aligned}
$$

Summing over all $T$ rounds, we obtain:

$$
\sum_{t=1}^T \mathbb{E}[v(x^t, d(s^t, a^t)) - v(x^t, a^t)] \leq T\delta = \frac{\mathrm{CSReg}(T)}{2}.
$$

- If the principal does not play according to $\sigma$ in any round, then the agent will switch to an algorithm with swap regret at most $\frac{\mathrm{CSReg}(T)}{2}$, so the total swap regret of the agent is at most:

$$
T\delta + \frac{\mathrm{CSReg}(T)}{2} \leq \mathrm{CSReg}(T),
$$

which proves that the agent's learning algorithm has swap regret at most $\mathrm{CSReg}(T)$.

The principal's average utility, when the principal uses strategy $\sigma$ and the agent uses the above no-swap-regret algorithm, is

$$
\begin{aligned}
\frac{1}{T}\sum_{t=1}^T \mathbb{E}[u(x^t, a^t)] &= \frac{1}{T}\sum_{t=1}^T \mathbb{E}_{h^{t-1}}\Big[\sum_{s \in S} \pi_s^t \sum_{a \in A} \rho^t(a|s)u(x_s^t, a)\Big] \\
&= \frac{1}{T}\sum_{t=1}^T \mathbb{E}_{h^{t-1}}\Big[U(\pi^t, \rho^t)\Big] \\
&\leq \frac{1}{T}\sum_{t=1}^T \mathbb{E}_{h^{t-1}}\Big[\sup_\pi \min_{\rho \in \mathcal{R}_\delta(\pi)} U(\pi, \rho)\Big] \quad \text{because } \rho^t \in \operatorname*{arg\,min}_{\rho \in \mathcal{R}_\delta(\pi^t)} U(\pi^t, \rho) \\
&= \sup_\pi \min_{\rho \in \mathcal{R}_\delta(\pi)} U(\pi, \rho) \\
&= \underline{\mathrm{OBJ}}^{\mathcal{R}}(\delta) \\
&= \underline{\mathrm{OBJ}}^{\mathcal{R}}\Big(\frac{\mathrm{CSReg}(T)}{2T}\Big).
\end{aligned}
$$

### E.3 PROOF OF THEOREM 3.4

*Proof.* Let $p_s^t = \Pr[s^t = s] = \mathbb{E}\big[\mathbb{1}[s^t = s]\big] = \mathbb{E}[\pi_s^t]$ be the probability that signal $s \in S$ is sent in round $t$. Let $p_{a|s}^t = \Pr[a^t = a \mid s^t = s]$ be the probability that the agent takes action $a$ conditioning on signal $s^t = s$ being sent in round $t$. Let $d : S \times A \to A$ be any deviation function for the agent.

The utility gain by deviation for the agent is upper bounded by the contextual swap-regret:

$$\frac{\text{CSReg}(T)}{T} \geq \frac{1}{T}\mathbb{E}\Big[\sum_{t=1}^{T}\Big(v(x^t, d(s^t, a^t)) - v(x^t, a^t)\Big)\Big] \tag{17}$$

$$= \frac{1}{T}\sum_{t=1}^{T}\sum_{s\in S}p_s^t\sum_{a\in A}p_{a|s}^t\mathbb{E}_{x_s^t|s^t=s}\Big[v(x_s^t, d(s,a)) - v(x_s^t, a)\Big]$$

$$= \frac{1}{T}\sum_{t=1}^{T}\sum_{s\in S}p_s^t\sum_{a\in A}p_{a|s}^t\Big(v(\mathbb{E}[x_s^t|s^t=s], d(s,a)) - v(\mathbb{E}[x_s^t|s^t=s], a)\Big) \qquad \text{by linearity of } v(\cdot, a)$$

$$= \sum_{s\in S}\sum_{a\in A}\frac{\sum_{j=1}^{T}p_s^j p_{a|s}^j}{T}\frac{1}{\sum_{j=1}^{T}p_s^j p_{a|s}^j}\sum_{t=1}^{T}p_s^t p_{a|s}^t\Big(v(\mathbb{E}[x_s^t|s^t=s], d(s,a)) - v(\mathbb{E}[x_s^t|s^t=s], a)\Big)$$

$$= \sum_{s\in S}\sum_{a\in A}\frac{\sum_{j=1}^{T}p_s^j p_{a|s}^j}{T}\Big[v\Big(\frac{\sum_{t=1}^{T}p_s^t p_{a|s}^t\mathbb{E}[x_s^t|s^t=s]}{\sum_{j=1}^{T}p_s^j p_{a|s}^j}, d(s,a)\Big) - v\Big(\frac{\sum_{t=1}^{T}p_s^t p_{a|s}^t\mathbb{E}[x_s^t|s^t=s]}{\sum_{j=1}^{T}p_s^j p_{a|s}^j}, a\Big)\Big].$$

Define $q_{s,a} = \frac{\sum_{j=1}^{T}p_s^j p_{a|s}^j}{T}$ and $y_{s,a} = \frac{\sum_{t=1}^{T}p_s^t p_{a|s}^t\mathbb{E}[x_s^t|s^t=s]}{\sum_{j=1}^{T}p_s^j p_{a|s}^j} \in \mathcal{X}$. Then the above is equal to

$$= \sum_{s\in S}\sum_{a\in A}q_{s,a}\Big[v(y_{s,a}, d(s,a)) - v(y_{s,a}, a)\Big]. \tag{18}$$

We note that $\sum_{s\in S}\sum_{a\in A}q_{s,a} = \frac{\sum_{j=1}^{T}\sum_{s\in S}\sum_{a\in A}p_s^j p_{a|s}^j}{T} = 1$, so $q$ is a probability distribution over $S \times A$. And note that

$$\sum_{s,a\in S\times A}q_{s,a}y_{s,a} = \sum_{s,a\in S\times A}\frac{1}{T}\sum_{t=1}^{T}p_s^t p_{a|s}^t\mathbb{E}[x_s^t|s^t=s] = \frac{1}{T}\sum_{t=1}^{T}\sum_{s\in S}p_s^t\mathbb{E}[x_s^t|s^t=s]$$

$$= \frac{1}{T}\sum_{t=1}^{T}\sum_{s\in S}\mathbb{E}\big[\mathbb{1}[s^t=s]x_s^t\big] = \frac{1}{T}\sum_{t=1}^{T}\mathbb{E}\big[\sum_{s\in S}\mathbb{1}[s^t=s]x_s^t\big] = \frac{1}{T}\sum_{t=1}^{T}\mathbb{E}\big[x^t\big]$$

$$= \frac{1}{T}\sum_{t=1}^{T}\mathbb{E}\big[\sum_{s\in S}\pi_s^t x_s^t\big] \in \mathcal{C} \qquad \text{because } \sum_{s\in S}\pi_s^t x_s^t \in \mathcal{C}.$$

This means that $\pi' = \{(q_{s,a}, y_{s,a})\}_{(s,a)\in S\times A}$ defines a valid principal strategy with the larger signal space $S \times A$. Then, we note that (18) is the difference between the agent's expected utility under principal strategy $\pi'$ when responding using strategy $d : S \times A \to A$ and using the strategy that maps signal $(s, a)$ to action $a$. And (18) is upper bounded by $\frac{\text{CSReg}(T)}{T}$ by (17):

$$(18) = V(\pi', d) - V(\pi', (s,a) \mapsto a) \leq \frac{\text{CSReg}(T)}{T}, \quad \forall d : S \times A \to A. \tag{19}$$

In particular, this holds when $d$ is the agent's best-responding strategy. This means that the agent strategy $(s, a) \mapsto a$ is a $(\frac{\text{CSReg}(T)}{T})$-best-response to $\pi'$. So, the principal's expected utility is upper bounded by the utility in the approximate-best-response model:

$$\frac{1}{T}\mathbb{E}\Big[\sum_{t=1}^{T}u(x^t, a^t)\Big] = \frac{1}{T}\sum_{t=1}^{T}\sum_{s\in S}p_s^t\sum_{a\in A}p_{a|s}^t v(\mathbb{E}[x_s^t|s^t=s], a)$$

$$= \sum_{s\in S}\sum_{a\in A}q_{s,a}u(y_{s,a}, a) = U(\pi', (s,a) \to a) \leq \overline{\text{OBJ}}^{\mathcal{R}}\Big(\frac{\text{CSReg}(T)}{T}\Big).$$

$\square$

### E.4 PROOF OF THEOREM 3.5

The instance has 2 states (A, B), 3 actions (L, M, R), uniform prior $\mu_0(A) = \mu_0(B) = 0.5$, with the following utility matrices (left for sender's, right for receiver's):

| $u(\omega, a)$ | L | M | R |
|---|---|---|---|
| A | 0 | $-2$ | $-2$ |
| B | 0 | 0 | 2 |

| $v(\omega, a)$ | L | M | R |
|---|---|---|---|
| A | $\sqrt{\gamma}$ | $-1$ | 0 |
| B | $-1$ | 1 | 0 |

**Claim 2.** *In this instance, the optimal sender utility $U^*$ in the classic BP model is $0$, and the approximate-best-response objective $\overline{\mathrm{OBJ}}^{\mathcal{R}}(\gamma) = O(\gamma)$.*

*Proof.* Recall that any signaling scheme decomposes the prior $\mu_0$ into multiple posteriors $\{\mu_s\}_{s \in S}$. If a posterior $\mu_s$ puts probability $> 0.5$ to state B, then the receiver will take action M, which gives the sender a utility $\leq 0$; if the posterior $\mu_s$ puts probability $\leq 0.5$ to state B, then no matter what action the receiver takes, the sender's expected utility on $\mu_s$ cannot be greater than 0. So, the sender's expected utility is $\leq 0$ under any signaling scheme. An optimal signaling scheme is to reveal no information (keep $\mu_s = \mu_0$); the receiver takes R and the sender gets utility 0.

This instance satisfies the assumptions of Theorem 4.2, so $\overline{\mathrm{OBJ}}^{\mathcal{R}}(\gamma) \leq U^* + O(\gamma) = O(\gamma)$. □

**Claim 3.** *By doing the following, the sender can obtain utility $\approx \frac{1}{2} - O(\sqrt{\gamma})$ if the receiver is $\gamma$-mean-based learning:*

- *in the first $T/2$ rounds: if the state is A, send signal 1; if the state is B, send 2.*

- *in the remaining $T/2$ rounds, switch the scheme: if the state is A, send 2; if state is B, send 1.*

*Proof.* In the first $T/2$ rounds, the receiver finds that signal 1 corresponds to state A so he will take action L with high probability when signal 1 is sent; signal 2 corresponds to B so he will take action M with high probability. In this phase, the sender obtains utility $\approx 0$ per round. At the end of this phase, for signal 1, the receiver accumulates utility $\approx \frac{T}{2} \frac{1}{2} \sqrt{\gamma} = \frac{T}{4} \sqrt{\gamma}$ for action L. For signal 2, the receiver accumulates utility $\approx \frac{T}{2} \frac{1}{2} \cdot 1 = \frac{T}{4}$ for action M.

In the remaining $T/2$ rounds, the following will happen:

- For signal 1, the receiver finds that the state is now B, so the utility of action L decreases by 1 every time signal 1 is sent. Because the utility of L accumulated in the first phase was $\approx \frac{T}{4} \sqrt{\gamma}$, after $\approx \frac{T}{4} \sqrt{\gamma}$ rounds in second phase the utility of L should decrease to below 0, and the receiver will no longer play L (with high probability) at signal 1. The receiver will not play M at signal 1 in most of the second phase either, because there are more A states than B states at signal 1 historically. So, the receiver will play action R most times, roughly $\frac{T}{4} - \frac{T}{4} \sqrt{\gamma}$ rounds. This gives the sender a total utility of $\approx (\frac{T}{4} - \frac{T}{4} \sqrt{\gamma}) \cdot 2 = \frac{T}{2} - O(T \sqrt{\gamma})$.

- For signal 2, the state is now A. But the receiver will continue to play action M in most times. This because: R has utility 0; L accumulated $\approx -\frac{T}{4}$ utility in the first phase, and only increases by $\sqrt{\gamma}$ per round in the second phase, so its accumulated utility is always negative; instead, M has accumulated $\frac{T}{4}$ utility in the first phase, and decreases by 1 every time signal 2 is sent in the second phase, so its utility is positive until near the end. So, the receiver will play M. This gives the sender utility 0.

Summing up, the sender obtains total utility $\approx \frac{T}{2} - O(T \sqrt{\gamma})$ in these two phases, which is $\frac{1}{2} - O(\sqrt{\gamma}) > 0$ per round in average. □

The above two claims together prove the theorem.

# F  MISSING PROOFS FROM SECTION 4

## F.1  PROOF OF CLAIM 1

If no $G > 0$ satisfies the claim, then there must exist an $a_0 \in A$ such that for all $x \in \mathcal{X}$, $v(a_0, \mu) - v(a', \mu) \leq 0$ for some $a' \in A \setminus \{a_0\}$. Namely,

$$\max_{x \in \mathcal{X}} \min_{a' \in A \setminus \{a_0\}} \big\{ v(x, a_0) - v(x, a') \big\} \leq 0.$$

Then, by the minimax theorem, we have

$$\min_{\alpha' \in \Delta(A \setminus \{a_0\})} \max_{x \in \mathcal{X}} \big\{ v(x, a_0) - v(x, \alpha') \big\} = \max_{x \in \mathcal{X}} \min_{a' \in A \setminus \{a_0\}} \big\{ v(x, a_0) - v(x, a') \big\} \leq 0.$$

This means that $a_0$ is weakly dominated by some mixed action $\alpha' \in \Delta(A \setminus \{a_0\})$, violating Assumption 1.

## F.2  PROOF OF EXAMPLE 4.1

We use the probability $\mu \in [0, 1]$ of the Good state to represent a belief (so the probability of Bad state is $1 - \mu$).

First, the sender's optimal utility when the receiver exactly best responds is $2\mu_0$:

$$U^* = 2\mu_0.$$

This is achieved by decomposing the prior $\mu_0$ into two posteriors $\mu_a = \frac{1}{2}$ and $\mu_b = 0$ with probability $2\mu_0$ and $1 - 2\mu_0$ respectively, with the receiver taking action $a$ under posterior $\mu_a$ and $b$ under $\mu_b$.

Then, consider any signaling scheme of the sender, $\pi = \{(\pi_s, \mu_s)\}_{s \in S}$, which is a decomposition of the prior $\mu_0$ into $|S|$ posteriors $\mu_s \in [0, 1]$ such that $\sum_{s \in S} \pi_s \mu_s = \mu_0$. Let $\rho : S \to \Delta(A)$ be a randomized strategy of the receiver, where $\rho(a|s)$ (and $\rho(b|s)$) denotes the probability that the receiver takes action $a$ (and $b$) under signal $s$. The sender's expected utility under $\pi$ and $\rho$ is:

$$U(\pi, \rho) = \sum_{s \in S} \pi_s \big[ \rho(a|s) \cdot 1 + \rho(b|s) \cdot 0 \big] = \sum_{s \in S} \pi_s \rho(a|s). \tag{20}$$

The receiver's utility when taking action $a$ at posterior $\mu_s$ is $\mu_s \cdot 1 + (1 - \mu_s) \cdot (-1) = 2\mu_s - 1$. So, the receiver's expected utility under $\pi$ and $\rho$ is

$$V(\pi, \rho) = \sum_{s \in S} \pi_s \big[ \rho(a|s) \cdot (2\mu_s - 1) + \rho(b|s) \cdot 0 \big] = \sum_{s \in S} \pi_s \rho(a|s)(2\mu_s - 1). \tag{21}$$

Clearly, the receiver's best response $\rho^*$ is to take action $a$ with certainty if and only if $\mu_s > \frac{1}{2}$, with expected utility

$$V(\pi, \rho^*) = \sum_{s : \mu_s > \frac{1}{2}} \pi_s(2\mu_s - 1). \tag{22}$$

To find $\underline{\text{OBJ}}^{\mathcal{R}}(\delta) = \sup_\pi \min_{\rho \in \mathcal{R}_\delta(\pi)} U(\pi, \rho)$, we fix any $\pi$ and solve the inner optimization problem (minimizing the sender's utility) regarding $\rho$:

$$\min_\rho \quad U(\pi, \rho) = \sum_{s \in S} \pi_s \rho(a|s)$$

$$\text{s.t.} \quad \rho \in \mathcal{R}_\delta(\pi) \iff \delta \geq V(\pi, \rho^*) - V(\pi, \rho)$$

$$= \sum_{s : \mu_s > \frac{1}{2}} \pi_s(2\mu_s - 1) - \sum_{s \in S} \pi_s \rho(a|s)(2\mu_s - 1).$$

Without loss of generality, we can assume that the solution $\rho$ satisfies $\rho(a|s) = 0$ whenever $\mu_s \leq \frac{1}{2}$ (if $\rho(a|s) > 0$ for some $\mu_s \leq \frac{1}{2}$, then making $\rho(a|s)$ to be 0 can decrease the objective

$\sum_{s \in S} \pi_s \rho(a|s)$ while still satisfying the constraint). So, the optimization problem can be simplified to:

$$\min_{\rho} \qquad U(\pi, \rho) = \sum_{s:\mu_s > \frac{1}{2}} \pi_s \rho(a|s)$$

$$\text{s.t.} \qquad \delta \geq \sum_{s:\mu_s > \frac{1}{2}} \pi_s(2\mu_s - 1) - \sum_{s:\mu_s > \frac{1}{2}} \pi_s \rho(a|s)(2\mu_s - 1)$$

$$= \sum_{s:\mu_s > \frac{1}{2}} \pi_s(2\mu_s - 1)(1 - \rho(a|s)),$$

$$\rho(a|s) \in [0, 1], \quad \forall s \in S : \mu_s > \frac{1}{2}.$$

We note that this is a fractional knapsack linear program, which has a greedy solution (e.g., (Korte & Vygen, 2012)): sort the signals with $\mu_s > \frac{1}{2}$ in increasing order of $2\mu_s - 1$ (equivalently, increasing order of $\mu_s$); label those signals by $s = 1, \ldots, n$; find the first position $k$ for which $\sum_{s=1}^{k} \pi_s(2\mu_s - 1) > \delta$:

$$k = \min \Big\{ j : \sum_{s=1}^{j} \pi_s(2\mu_s - 1) > \delta \Big\};$$

then, an optimal solution $\rho$ is given by:

$$\begin{cases} \rho(a|s) = 0 & \text{for } s = 1, \ldots, k - 1; \\ \rho(a|k) = 1 - \frac{\delta - \sum_{s=1}^{k-1} \pi_s(2\mu_s - 1)}{\pi_k(2\mu_k - 1)} & \text{for } s = k; \\ \rho(a|s) = 1 & \text{for } s = k + 1, \ldots, n. \end{cases}$$

The objective value (sender's expected utility) of the above solution $\rho$ is

$$U(\pi, \rho) = \sum_{s:\mu_s > \frac{1}{2}} \pi_s \rho(a|s)$$

$$= \pi_k \Big( 1 - \frac{\delta - \sum_{s=1}^{k-1} \pi_s(2\mu_s - 1)}{\pi_k(2\mu_k - 1)} \Big) + \sum_{s=k+1}^{n} \pi_s$$

$$= \sum_{s=k}^{n} \pi_s - \frac{\delta}{2\mu_k - 1} + \sum_{s=1}^{k-1} \frac{\pi_s(2\mu_s - 1)}{2\mu_k - 1}.$$

Since the signaling scheme $\pi$ must satisfy $\sum_{s \in S} \pi_s \mu_s = \mu_0$, we have

$$\mu_0 = \sum_{s \in S} \pi_s \mu_s \geq \sum_{s=1}^{n} \pi_s \mu_s = \sum_{s=1}^{k-1} \pi_s \mu_s + \sum_{s=k}^{n} \pi_s \mu_s \geq \sum_{s=1}^{k-1} \pi_s \mu_s + \sum_{s=k}^{n} \pi_s \mu_k$$

$$\implies \sum_{s=k}^{n} \pi_s \leq \frac{\mu_0 - \sum_{s=1}^{k-1} \pi_s \mu_s}{\mu_k}.$$

So,

$$U(\pi, \rho) \leq \frac{\mu_0 - \sum_{s=1}^{k-1} \pi_s \mu_s}{\mu_k} - \frac{\delta}{2\mu_k - 1} + \sum_{s=1}^{k-1} \frac{\pi_s(2\mu_s - 1)}{2\mu_k - 1}$$

$$= \frac{\mu_0}{\mu_k} - \frac{\delta}{2\mu_k - 1} + \sum_{s=1}^{k-1} \pi_s \Big( \frac{2\mu_s - 1}{2\mu_k - 1} - \frac{\mu_s}{\mu_k} \Big).$$

Since $\frac{2\mu_s - 1}{2\mu_k - 1} - \frac{\mu_s}{\mu_k} = \frac{\mu_s - \mu_k}{(2\mu_s - 1)\mu_k} \leq 0$ for any $s \leq k - 1$, we get

$$U(\pi, \rho) \leq \frac{\mu_0}{\mu_k} - \frac{\delta}{2\mu_k - 1} = f(\mu_k).$$

We find the maximal value of $f(\mu_k) = \frac{\mu_0}{\mu_k} - \frac{\delta}{2\mu_k - 1}$. Take its derivative:

$$f'(\mu_k) = -\frac{\mu_0}{\mu_k^2} + \frac{2\delta}{(2\mu_k - 1)^2} = \frac{\left[(\sqrt{2\delta} + 2\sqrt{\mu_0})\mu_k - \sqrt{\mu_0}\right] \cdot \left[(\sqrt{2\delta} - 2\sqrt{\mu_0})\mu_k + \sqrt{\mu_0}\right]}{\mu_k^2(2\mu_k - 1)^2},$$

which has two roots $\frac{\sqrt{\mu_0}}{\sqrt{2\delta} + 2\sqrt{\mu_0}} < \frac{1}{2}$ and $\frac{\sqrt{\mu_0}}{2\sqrt{\mu_0} - \sqrt{2\delta}} \in (\frac{1}{2}, 1)$ when $0 < \delta < \frac{\mu_0}{2}$. So, $f(x)$ is increasing in $[\frac{1}{2}, \frac{\sqrt{\mu_0}}{2\sqrt{\mu_0} - \sqrt{2\delta}})$ and decreasing in $(\frac{\sqrt{\mu_0}}{2\sqrt{\mu_0} - \sqrt{2\delta}}, 1]$. Since $\mu_k > \frac{1}{2}$, $f(\mu_k)$ is maximized at $\mu_k = \frac{\sqrt{\mu_0}}{2\sqrt{\mu_0} - \sqrt{2\delta}}$. This implies

$$U(\pi, \rho) \le f\left(\frac{\sqrt{\mu_0}}{2\sqrt{\mu_0} - \sqrt{2\delta}}\right) = \frac{\mu_0}{\sqrt{\mu_0}}(2\sqrt{\mu_0} - \sqrt{2\delta}) - \frac{\delta}{2\frac{\sqrt{\mu_0}}{2\sqrt{\mu_0} - \sqrt{2\delta}} - 1} = 2\mu_0 - 2\sqrt{2\mu_0\delta} + \delta.$$

This holds for any $\pi$. So, $\underline{\text{OBJ}}^{\mathcal{R}}(\delta) = \sup_\pi \min_{\rho \in \mathcal{R}_\delta(\pi)} U(\pi, \rho) \le U^* - 2\sqrt{2\mu_0\delta} + \delta = U^* - \Omega(\sqrt{\delta})$.

### F.3    PROOF OF THEOREMS 4.1 AND 4.2

**Lower bounds on $\underline{\text{OBJ}}^{\mathcal{D}}(\delta)$ and upper bounds on $\overline{\text{OBJ}}^{\mathcal{R}}(\delta)$.**    First, we prove the lower bounds on $\underline{\text{OBJ}}^{\mathcal{D}}(\delta)$ and the upper bounds on $\overline{\text{OBJ}}^{\mathcal{R}}(\delta)$ in Theorems 4.1 and 4.2, given by the following two lemmas:

**Lemma F.1.** *In an unconstrained generalized principal-agent problem,* $\underline{\text{OBJ}}^{\mathcal{D}}(\delta) \ge U^* - \text{diam}(\mathcal{X})L\frac{\delta}{G}$.

*With the constraint $\sum_{s \in S} \pi_s x_s \in \mathcal{C}$,* $\underline{\text{OBJ}}^{\mathcal{D}}(\delta) \ge U^* - \left(\text{diam}(\mathcal{X})L + 2B\frac{\text{diam}(\mathcal{X})}{\text{dist}(\mathcal{C}, \partial\mathcal{X})}\right)\frac{\delta}{G}$.

**Lemma F.2.** *In an unconstrained generalized principal-agent problem,* $\overline{\text{OBJ}}^{\mathcal{R}}(\delta) \le U^* + \text{diam}(\mathcal{X})L\frac{\delta}{G}$.

*With the constraint $\sum_{s \in S} \pi_s x_s \in \mathcal{C}$,* $\overline{\text{OBJ}}^{\mathcal{R}}(\delta) \le U^* + \left(\text{diam}(\mathcal{X})L + 2B\frac{\text{diam}(\mathcal{X})}{\text{dist}(\mathcal{C}, \partial\mathcal{X})}\right)\frac{\delta}{G}$.

The proofs of Lemmas F.1 and F.2 are similar and given in Appendix F.4 and F.5. The main idea to prove Lemma F.2 is the following. Let $(\pi, \rho)$ be any pair of principal's strategy and agent's $\delta$-best-responding strategy. We perturb the principal's strategy $\pi$ slightly to be a strategy $\pi'$ for which $\rho$ is *exactly* best-responding (such a perturbation is possible due to Assumption 1). Since $\rho$ is best-responding to $\pi'$, the pair $(\pi', \rho)$ cannot give the principal a higher utility than $U^*$ (which is the optimal principal utility under the best-response model). This means that the original pair $(\pi, \rho)$ cannot give the principal a utility much higher than $U^*$, implying an upper bound on $\overline{\text{OBJ}}^{\mathcal{R}}(\delta)$.

**Upper bounds on $\overline{\text{OBJ}}^{\mathcal{R}}(\delta)$ imply upper bounds on $\overline{\text{OBJ}}^{\mathcal{D}}(\delta)$.**    Then, because $\overline{\text{OBJ}}^{\mathcal{D}}(\delta) \le \overline{\text{OBJ}}^{\mathcal{R}}(\delta)$, we immediately obtain the upper bounds on $\overline{\text{OBJ}}^{\mathcal{D}}(\delta)$ in the two theorems.

**Lower bounds for $\underline{\text{OBJ}}^{\mathcal{D}}(\delta)$ imply lower bounds for $\underline{\text{OBJ}}^{\mathcal{R}}(\delta)$**    Finally, we show that the lower bounds for $\underline{\text{OBJ}}^{\mathcal{D}}(\delta)$ imply the lower bounds for $\underline{\text{OBJ}}^{\mathcal{R}}(\delta)$, using the following lemma:

**Lemma F.3.** *For any $\delta \ge 0, \Delta > 0$,* $\underline{\text{OBJ}}^{\mathcal{R}}(\delta) \ge \underline{\text{OBJ}}^{\mathcal{D}}(\Delta) - \frac{2B\delta}{\Delta}$.

The proof of this lemma is in Appendix F.6.

Using Lemma F.3 with $\Delta = \sqrt{\frac{2BG\delta}{\text{diam}(\mathcal{X})L}}$ and the lower bound for $\underline{\text{OBJ}}^{\mathcal{D}}(\Delta)$ in Lemma F.1 for the unconstrained case, we obtain:

$$\underline{\text{OBJ}}^{\mathcal{R}}(\delta) \ge \underline{\text{OBJ}}^{\mathcal{D}}(\Delta) - \frac{2B\delta}{\Delta} \ge U^* - \text{diam}(\mathcal{X})L\frac{\Delta}{G} - \frac{2B\delta}{\Delta} = U^* - 2\sqrt{\frac{2BL}{G}\text{diam}(\mathcal{X})\delta},$$

which gives the lower bound for $\underline{\text{OBJ}}^{\mathcal{R}}(\delta)$ in Theorem 4.1.

Using Lemma F.3 with $\Delta = \sqrt{\frac{2BG\delta}{L\mathrm{diam}(\mathcal{X})+2B\frac{\mathrm{diam}(\mathcal{X})}{\mathrm{dist}(\mathcal{C},\partial\mathcal{X})}}}$ and the lower bound for $\underline{\mathrm{OBJ}}^{\mathcal{D}}(\Delta)$ in Lemma F.1 for the constrained case, we obtain:

$$\underline{\mathrm{OBJ}}^{\mathcal{R}}(\delta) \geq \underline{\mathrm{OBJ}}^{\mathcal{D}}(\Delta) - \frac{2B\delta}{\Delta} \geq U^* - \left(\mathrm{diam}(\mathcal{X})L + 2B\frac{\mathrm{diam}(\mathcal{X})}{\mathrm{dist}(\mathcal{C},\partial\mathcal{X})}\right)\frac{\Delta}{G} - \frac{2B\delta}{\Delta}$$

$$= U^* - 2\sqrt{\frac{2B}{G}\left(\mathrm{diam}(\mathcal{X})L + 2B\frac{\mathrm{diam}(\mathcal{X})}{\mathrm{dist}(\mathcal{C},\partial\mathcal{X})}\right)\delta}.$$

This proves the lower bound for $\underline{\mathrm{OBJ}}^{\mathcal{R}}(\delta)$ in Theorem 4.2.

### F.4 PROOF OF LEMMA F.1

Let $(\pi, \rho)$ be a pair of principal strategy and agent strategy that achieves the optimal principal utility with an exactly-best-responding agent, namely, $U(\pi, \rho) = U^*$. Without loss of generality $\rho$ can be assumed to be deterministic, $\rho : S \to A$. The strategy $\pi$ consists of pairs $\{(\pi_s, x_s)\}_{s \in S}$ that satisfy

$$\sum_{s \in S} \pi_s x_s =: \mu_0 \in \mathcal{C}, \tag{23}$$

and the action $a = \rho(s)$ is optimal for the agent with respect to $x_s$. We will construct another principal strategy $\pi'$ such that, even if the agent chooses the worst $\delta$-best-responding strategy to $\pi'$, the principal can still obtain utility arbitrarily close to $U^* - \left(L\mathrm{diam}(\mathcal{X}; \ell_1) + 2B\frac{\mathrm{diam}(\mathcal{X})}{\mathrm{dist}(\mathcal{C},\partial\mathcal{X})}\right)\frac{\delta}{G}$.

To construct $\pi'$ we do the following: For each signal $s \in S$, with corresponding action $a = \rho(s)$, by Claim 1 there exists $y_a \in \mathcal{X}$ such that $v(y_a, a) - v(y_a, a') \geq G$ for any $a' \neq a$. Let $\theta = \frac{\delta}{G} + \varepsilon \in [0, 1]$ for arbitrarily small $\varepsilon > 0$, and let $\tilde{x}_s$ be the convex combination of $x_s$ and $y_{\rho(s)}$ with weights $1 - \theta, \theta$:

$$\tilde{x}_s = (1 - \theta)x_s + \theta y_{\rho(s)}. \tag{24}$$

We note that $a = \rho(s)$ is the agent's optimal action for $\tilde{x}_s$ and moreover it is better than any other action $a' \neq a$ by more than $\delta$:

$$v(\tilde{x}_s, a) - v(\tilde{x}_s, a') = (1 - \theta)\big[\underbrace{v(x_s, a) - v(x_s, a')}_{\geq 0 \text{ because } a = \rho(s) \text{ is optimal for } x_s}\big] + \theta\big[\underbrace{v(y_a, a) - v(y_a, a')}_{\geq G \text{ by our choice of } y_a}\big]$$

$$\geq 0 + \theta G > \frac{\delta}{G}G = \delta. \tag{25}$$

Let $\mu'$ be the convex combination of $\{\tilde{x}_s\}_{s \in S}$ with weights $\{\pi_s\}_{s \in S}$:

$$\mu' = \sum_{s \in S} \pi_s \tilde{x}_s. \tag{26}$$

Note that $\mu'$ might not satisfy the constraint $\mu' \in \mathcal{C}$. So, we want to find another vector $z \in \mathcal{X}$ and a coefficient $\eta \in [0, 1]$ such that

$$(1 - \eta)\mu' + \eta z \in \mathcal{C}. \tag{27}$$

(If $\mu'$ already satisfies $\mu' \in \mathcal{C}$, then let $\eta = 0$.) To do this, we consider the ray starting from $\mu'$ pointing towards $\mu_0$: $\{\mu' + t(\mu_0 - \mu') \mid t \geq 0\}$. Let $z$ be the intersection of the ray with the boundary of $\mathcal{X}$:

$$z = \mu' + t^*(\mu_0 - \mu'), \qquad t^* = \arg\max\{t \geq 0 \mid \mu' + t(\mu_0 - \mu') \in \mathcal{X}\}.$$

Then, rearranging $z = \mu' + t^*(\mu_0 - \mu')$, we get

$$\tfrac{1}{t^*}(z - \mu') = \mu_0 - \mu' \quad \Longleftrightarrow \quad (1 - \tfrac{1}{t^*})\mu' + \tfrac{1}{t^*}z = \mu_0 \in \mathcal{C},$$

which satisfies (27) with $\eta = \frac{1}{t^*}$. We then give an upper bound on $\eta = \frac{1}{t^*}$:

**Claim 4.** $\eta = \frac{1}{t^*} \leq \frac{\mathrm{diam}(\mathcal{X})}{\mathrm{dist}(\mathcal{C},\partial\mathcal{X})}\theta.$

*Proof.* On the one hand,

$$\|\mu_0 - \mu'\| = \Big\|\sum_{s \in S} \pi_s x_s - \sum_{s \in S} \pi_s \tilde{x}_s\Big\| = \Big\|\sum_{s \in S} \pi_s \theta(y_{\rho(s)} - x_s)\Big\|$$

$$\leq \theta \sum_{s \in S} \pi_s \big\|y_{\rho(s)} - x_s\big\| \leq \theta \sum_{s \in S} \pi_s \cdot \mathrm{diam}(\mathcal{X}) = \theta \cdot \mathrm{diam}(\mathcal{X}).$$

On the other hand, because $z - \mu'$ and $\mu_0 - \mu'$ are in the same direction, we have

$$\|z - \mu'\| = \|z - \mu_0\| + \|\mu_0 - \mu'\| \geq \|z - \mu_0\| \geq \mathrm{dist}(\mathcal{C}, \partial\mathcal{X})$$

because $\mu_0$ is in $\mathcal{C}$ and $z$ is on the boundary of $\mathcal{X}$. Therefore, $\eta = \frac{1}{t^*} = \frac{\|\mu_0 - \mu'\|}{\|z - \mu'\|} \leq \frac{\mathrm{diam}(\mathcal{X})}{\mathrm{dist}(\mathcal{C}, \partial\mathcal{X})}\theta$. $\quad\square$

The convex combinations (27) (26) define a new principal strategy $\pi'$ with $|S|+1$ signals, consisting of $\tilde{x}_s$ with probability $(1 - \eta)\pi_s$ and $z$ with probability $\eta$, satisfying $\sum_{s \in S}(1 - \eta)\pi_s\tilde{x}_s + \eta z = \mu_0 \in \mathcal{C}$. Consider the agent's worst (for the principal) $\delta$-best-responding strategies $\rho'$ to $\pi'$:

$$\rho' \in \underset{\rho \in \mathcal{D}_\delta(\pi')}{\arg\min}\, U(\pi', \rho).$$

We note that $\rho'(\tilde{x}_s)$ must be equal to $\rho(s)$ for each $s \in S$. This is because $a = \rho(s)$ is strictly better than any other action $a' \neq a$ by a margin of $\delta$ (25), so $a$ is the only $\delta$-optimal action for $\tilde{x}_s$.

Then, the principal's expected utility under $\pi'$ and $\rho'$ is

$$
\begin{aligned}
U(\pi', \rho') &\overset{(27),(26)}{=} (1 - \eta)\sum_{s \in S}\pi_s u(\tilde{x}_s, \rho'(\tilde{x}_s)) \;+\; \eta u(z, \rho'(z)) \\
&\geq (1 - \eta)\sum_{s \in S}\pi_s u(\tilde{x}_s, \rho(s)) \;-\; \eta B \\
&\geq (1 - \eta)\sum_{s \in S}\pi_s\Big(u(x_s, \rho(s)) - L\underbrace{\|\tilde{x}_s - x_s\|}_{=\|\theta(y_{\rho(s)} - x_s)\| \leq \theta\mathrm{diam}(\mathcal{X})}\Big) \;-\; \eta B \\
&\geq (1 - \eta)U(\pi, \rho) - L\theta\mathrm{diam}(\mathcal{X}) - \eta B \\
&\geq U(\pi, \rho) - L\theta\mathrm{diam}(\mathcal{X}) - 2\eta B \\
(\text{Claim 4}) \quad &\geq U(\pi, \rho) - L\theta\mathrm{diam}(\mathcal{X}) - 2B\frac{\mathrm{diam}(\mathcal{X})}{\mathrm{dist}(\mathcal{C}, \partial\mathcal{X})}\theta \\
&= U(\pi, \rho) - \Big(L\mathrm{diam}(\mathcal{X}) + 2B\frac{\mathrm{diam}(\mathcal{X})}{\mathrm{dist}(\mathcal{C}, \partial\mathcal{X})}\Big)\big(\tfrac{\delta}{G} + \varepsilon\big) \\
&= U^* - \Big(L\mathrm{diam}(\mathcal{X}) + 2B\frac{\mathrm{diam}(\mathcal{X})}{\mathrm{dist}(\mathcal{C}, \partial\mathcal{X})}\Big)\tfrac{\delta}{G} - O(\varepsilon).
\end{aligned}
$$

So, we conclude that

$$
\underline{\mathrm{OBJ}}^{\mathcal{P}}(\delta) = \sup_{\pi}\min_{\rho \in \mathcal{D}_\delta(\pi)}U(\pi, \rho) \geq \min_{\rho \in \mathcal{D}_\delta(\pi')}U(\pi', \rho)
$$
$$
= U(\pi', \rho') \geq U^* - \Big(L\mathrm{diam}(\mathcal{X}) + 2B\frac{\mathrm{diam}(\mathcal{X})}{\mathrm{dist}(\mathcal{C}, \partial\mathcal{X})}\Big)\tfrac{\delta}{G} - O(\varepsilon).
$$

Letting $\varepsilon \to 0$ finishes the proof for the case with the constraint $\sum_{s \in S}\pi_s x_s \in \mathcal{C}$.

The case without $\sum_{s \in S}\pi_s x_s \in \mathcal{C}$ is proved by letting $\eta = 0$ in the above argument.

### F.5 Proof of Lemma F.2

Let $\pi$ be a principal strategy and $\rho \in \mathcal{R}_\delta(\pi)$ be a $\delta$-best-responding randomized strategy of the agent. The principal strategy $\pi$ consists of pairs $\{(\pi_s, x_s)\}_{s \in S}$ with

$$\sum_{s \in S}\pi_s x_s =: \mu_0 \in \mathcal{C}. \tag{28}$$

At signal $s$, the agent takes action $a$ with probability $\rho(a|s)$. Let $\delta_{s,a}$ be the "suboptimality" of action $a$ with respect to $x_s$:

$$\delta_{s,a} = \max_{a' \in A}\big\{v(x_s, a') - v(x_s, a)\big\}. \tag{29}$$

By Claim 1, for action $a$ there exists $y_a \in \mathcal{X}$ such that $v(y_a, a) - v(y_a, a') \geq G$ for any $a' \neq a$. Let $\theta_{s,a} = \frac{\delta_{s,a}}{G + \delta_{s,a}} \in [0, 1]$ and let $\tilde{x}_{s,a}$ be the convex combination of $x_s$ and $y_a$ with weights $1 - \theta_{s,a}$ and $\theta_{s,a}$:

$$\tilde{x}_{s,a} = (1 - \theta_{s,a})x_s + \theta_{s,a}y_a. \tag{30}$$

**Claim 5.** *We have two useful claims regarding $\tilde{x}_{s,a}$ and $\theta_{s,a}$:*

*(1) $a$ is an optimal action for the agent with respect to $\tilde{x}_{s,a}$: $v(\tilde{x}_{s,a}, a) - v(\tilde{x}_{s,a}, a') \geq 0, \forall a' \in A$.*

*(2) $\sum_{s \in S} \sum_{a \in A} \pi_s \rho(a|s) \theta_{s,a} \leq \frac{\delta}{G}$.*

*Proof.* (1) For any $a' \neq a$, by the definition of $\tilde{x}_{s,a}$ and $\theta_{s,a}$,

$$
\begin{aligned}
v(\tilde{x}_{s,a}, a) - v(\tilde{x}_{s,a}, a') &= (1 - \theta_{s,a})\big[v(x_s, a) - v(x_s, a')\big] + \theta_{s,a}\big[v(y_a, a) - v(y_a, a')\big] \\
&\geq (1 - \theta_{s,a})(-\delta_{s,a}) + \theta_{s,a}G = \frac{G}{G + \delta_{s,a}}(-\delta_{s,a}) + \frac{\delta_{s,a}}{G + \delta_{s,a}}G = 0.
\end{aligned}
$$

(2) By the condition that $\rho$ is a $\delta$-best-response to $\pi$, we have

$$
\begin{aligned}
\delta &\geq \max_{\rho^*: S \to A} V(\pi, \rho^*) - V(\pi, \rho) = \sum_{s \in S} \pi_s \Big( \max_{a' \in A} \{v(x_s, a')\} - \sum_{a \in A} \rho(a|s) v(x_s, a) \Big) \\
&= \sum_{s \in S} \sum_{a \in A} \pi_s \rho(a|s) \max_{a' \in A} \{v(x_s, a') - v(x_s, a)\} = \sum_{s \in S} \sum_{a \in A} \pi_s \rho(a|s) \delta_{s,a}.
\end{aligned}
$$

So, $\sum_{s \in S} \sum_{a \in A} \pi_s \rho(a|s) \theta_{s,a} = \sum_{s \in S} \sum_{a \in A} \pi_s \rho(a|s) \frac{\delta_{s,a}}{G + \delta_{s,a}} \leq \sum_{s \in S} \sum_{a \in A} \pi_s \rho(a|s) \frac{\delta_{s,a}}{G} \leq \frac{\delta}{G}$. $\qquad \square$

We let $\mu'$ be the convex combination of $\{\tilde{x}_{s,a}\}_{s,a \in S \times A}$ with weights $\{\pi_s \rho(a|s)\}_{s,a \in S \times A}$:

$$
\mu' = \sum_{s,a \in S \times A} \pi_s \rho(a|s) \tilde{x}_{s,a}. \tag{31}
$$

Note that $\mu'$ might not satisfy the constraint $\mu' \in \mathcal{C}$. So, we want to find another vector $z \in \mathcal{X}$ and a coefficient $\eta \in [0, 1]$ such that

$$
(1 - \eta)\mu' + \eta z \in \mathcal{C}. \tag{32}
$$

(If $\mu'$ already satisfies $\mu' \in \mathcal{C}$, then let $\eta = 0$.) To do this, we consider the ray pointing from $\mu'$ to $\mu_0$: $\{\mu' + t(\mu_0 - \mu') \mid t \geq 0\}$. Let $z$ be the intersection of the ray with the boundary of $\mathcal{X}$:

$$
z = \mu' + t^*(\mu_0 - \mu'), \qquad t^* = \arg\max\{t \geq 0 \mid \mu' + t(\mu_0 - \mu') \in \mathcal{X}\}.
$$

Then, rearranging $z = \mu' + t^*(\mu_0 - \mu')$, we get

$$
\frac{1}{t^*}(z - \mu') = \mu_0 - \mu' \quad \Longleftrightarrow \quad (1 - \frac{1}{t^*})\mu' + \frac{1}{t^*}z = \mu_0 \in \mathcal{C},
$$

which satisfies (32) with $\eta = \frac{1}{t^*}$. We then give an upper bound on $\eta = \frac{1}{t^*}$:

**Claim 6.** $\eta = \frac{1}{t^*} \leq \frac{\mathrm{diam}(\mathcal{X})}{\mathrm{dist}(\mathcal{C}, \partial \mathcal{X})} \frac{\delta}{G}$.

*Proof.* On the one hand,

$$
\begin{aligned}
\|\mu_0 - \mu'\| &= \Big\| \sum_{s \in S} \pi_s x_s - \sum_{s \in S} \sum_{a \in A} \pi_s \rho(a|s) \tilde{x}_{s,a} \Big\| = \Big\| \sum_{s \in S} \sum_{a \in A} \pi_s \rho(a|s) \theta_{s,a}(y_a - x_s) \Big\| \\
&\leq \sum_{s \in S} \sum_{a \in A} \pi_s \rho(a|s) \theta_{s,a} \|y_a - x_s\| \leq \sum_{s \in S} \sum_{a \in A} \pi_s \rho(a|s) \theta_{s,a} \mathrm{diam}(\mathcal{X}) \overset{\text{Claim 5}}{\leq} \mathrm{diam}(\mathcal{X}) \frac{\delta}{G}.
\end{aligned}
$$

On the other hand, because $z - \mu'$ and $\mu_0 - \mu'$ are in the same direction, we have

$$
\|z - \mu'\| = \|z - \mu_0\| + \|\mu_0 - \mu'\| \geq \|z - \mu_0\| \geq \mathrm{dist}(\mathcal{C}, \partial \mathcal{X})
$$

because $\mu_0$ is in $\mathcal{C}$ and $z$ is on the boundary of $\mathcal{X}$. Therefore, $\eta = \frac{1}{t^*} = \frac{\|\mu_0 - \mu'\|}{\|z - \mu'\|} \leq \frac{\mathrm{diam}(\mathcal{X})}{\mathrm{dist}(\mathcal{C}, \partial \mathcal{X})} \frac{\delta}{G}$.

$\qquad \square$

The convex combinations (32) (31) define a new principal strategy $\pi'$ (with $|S| \times |A| + 1$ signals) consisting of $\tilde{x}_{s,a}$ with probability $(1 - \eta)\pi_s\rho(a|s)$ and $z$ with probability $\eta$. Consider the following deterministic agent strategy $\rho'$ in response to $\pi'$: for $\tilde{x}_{s,a}$, take action $\rho'(\tilde{x}_{s,a}) = a$; for $z$, take any action that is optimal for $z$. We note that $\rho'$ is a best-response to $\pi'$, $\rho' \in \mathcal{R}_0(\pi')$, because, according to Claim 5, $a$ is an optimal action with respect to $\tilde{x}_{s,a}$.

Then, consider the principal's utility under $\pi'$ and $\rho'$:

$$U(\pi', \rho') \overset{(32),(31)}{=} (1 - \eta) \sum_{s \in S} \sum_{a \in A} \pi_s \rho(a|s) u(\tilde{x}_{s,a}, \rho'(\tilde{x}_{s,a})) + \eta u(z, \rho'(z))$$

$$\geq (1 - \eta) \sum_{s \in S} \sum_{a \in A} \pi_s \rho(a|s) u(\tilde{x}_{s,a}, a) - \eta B$$

$$\geq (1 - \eta) \sum_{s \in S} \sum_{a \in A} \pi_s \rho(a|s) \Big( u(x_s, a) - L \underbrace{\|\tilde{x}_s - x_s\|}_{=\|\theta_{s,a}(y_a - x_s)\| \leq \theta_{s,a} \mathrm{diam}(\mathcal{X})} \Big) - \eta B$$

$$\geq (1 - \eta) U(\pi, \rho) - L \mathrm{diam}(\mathcal{X}) \sum_{s \in S} \sum_{a \in A} \pi_s \rho(a|s) \theta_{s,a} - \eta B$$

$$\text{(Claim 5)} \geq U(\pi, \rho) - L \mathrm{diam}(\mathcal{X}) \tfrac{\delta}{G} - 2\eta B$$

$$\text{(Claim 6)} \geq U(\pi, \rho) - \big( L \mathrm{diam}(\mathcal{X}) + 2B \tfrac{\mathrm{diam}(\mathcal{X})}{\mathrm{dist}(\mathcal{C}, \partial \mathcal{X})} \big) \tfrac{\delta}{G}.$$

Rearranging, $U(\pi, \rho) \leq U(\pi', \rho') + \big( L \mathrm{diam}(\mathcal{X}) + 2B \tfrac{\mathrm{diam}(\mathcal{X})}{\mathrm{dist}(\mathcal{C}, \partial \mathcal{X})} \big) \tfrac{\delta}{G}$. Note that this argument holds for any pair $(\pi, \rho)$ that satisfies $\rho \in \mathcal{R}_\delta(\pi)$. And recall that $\rho' \in \mathcal{R}_0(\pi')$. So, we conclude that

$$\overline{\mathrm{OBJ}}^{\mathcal{R}}(\delta) = \max_{\pi} \max_{\rho \in \mathcal{R}_\delta(\pi)} U(\pi, \rho) \leq \max_{\pi'} \max_{\rho' \in \mathcal{R}_0(\pi)} U(\pi', \rho') + \big( L \mathrm{diam}(\mathcal{X}; \ell_1) + 2B \tfrac{\mathrm{diam}(\mathcal{X})}{\mathrm{dist}(\mathcal{C}, \partial \mathcal{X})} \big) \tfrac{\delta}{G}$$

$$= U^* + \big( L \mathrm{diam}(\mathcal{X}; \ell_1) + 2B \tfrac{\mathrm{diam}(\mathcal{X})}{\mathrm{dist}(\mathcal{C}, \partial \mathcal{X})} \big) \tfrac{\delta}{G}.$$

This proves the case with the constraint $\sum_{s \in S} \pi_s x_s \in \mathcal{C}$.

The case without $\sum_{s \in S} \pi_s x_s \in \mathcal{C}$ is proved by letting $\eta = 0$ in the above argument.

### F.6   PROOF OF LEMMA F.3

Let $A_\Delta(x) = \big\{ a \in A \mid v(x, a) \geq v(x, a') - \Delta, \forall a' \in A \big\}$ be the set of $\Delta$-optimal actions of the agent in response to principal decision $x \in \mathcal{X}$. The proof of Lemma F.3 uses another lemma that relates the principal utility under a randomized $\delta$-best-responding agent strategy $\rho \in \mathcal{R}_\delta(\pi)$ and that under an agent strategy $\rho'$ that only randomizes over $A_\Delta(x_s)$.

**Lemma F.4.** *Let $\pi = \{(\pi_s, x_s)\}_{s \in S}$ be a principal strategy and $\rho \in \mathcal{R}_\delta(\pi)$ be a randomized $\delta$-best-response to $\pi$. For any $\Delta > 0$, there exists an agent strategy $\rho' : s \mapsto \Delta(A_\Delta(x_s))$ that randomizes over $\Delta$-optimal actions only for each $x_s$, such that the principal's utility under $\rho'$ and $\rho$ satisfies: $\big| U(\pi, \rho') - U(\pi, \rho) \big| \leq \tfrac{2B\delta}{\Delta}$.*

*Proof.* Let $a_s^* = \max_{a \in A} v(x_s, a)$ be the agent's optimal action for $x_s$. Let $\overline{A_\Delta(x_s)} = A \setminus A_\Delta(x_s)$ be the set of actions that are not $\Delta$-optimal for $x_s$. By the definition that $\rho \in \mathcal{R}_\delta(\pi)$ is a $\delta$-best-response to $\pi$, we have

$$\delta \geq \sum_{s \in S} \pi_s \Big[ v(x_s, a_s^*) - \sum_{a \in A} \rho(a|s) v(x_s, a) \Big]$$

$$= \sum_{s \in S} \pi_s \Big( \sum_{a \in A_\Delta(x_s)} \rho(a|s) \big[ \underbrace{v(x_s, a_s^*) - v(x_s, a)}_{\geq 0} \big] + \sum_{a \in \overline{A_\Delta(x_s)}} \rho(a|s) \big[ \underbrace{v(x_s, a_s^*) - v(x_s, a)}_{> \Delta} \big] \Big)$$

$$\geq 0 + \Delta \sum_{s \in S} \pi_s \sum_{a \in \overline{A_\Delta(x_s)}} \rho(a|s)$$

$$= \Delta \sum_{s \in S} \pi_s \rho(\overline{A_\Delta(x_s)} \,|\, s).$$

Rearranging,

$$\sum_{s \in S} \pi_s \rho(\overline{A_\Delta(x_s)} \mid s) \leq \frac{\delta}{\Delta}. \tag{33}$$

Then, we consider the randomized strategy $\rho'$ that, for each $s$, chooses each action $a \in A_\Delta(x_s)$ with the conditional probability that $\rho$ chooses $a$ given $a \in A_\Delta(x_s)$:

$$\rho'(a \mid s) = \frac{\rho(a \mid s)}{\rho(A_\Delta(x_s) \mid s)}.$$

The sender's utility under $\rho'$ is:

$$U(\pi, \rho') = \sum_{s \in S} \pi_s \sum_{a \in A_\Delta(x_s)} \frac{\rho(a \mid s)}{\rho(A_\Delta(x_s) \mid s)} u(x_s, a).$$

The sender's utility under $\rho$ is

$$U(\pi, \rho) = \sum_{s \in S} \pi_s \sum_{a \in A_\Delta(x_s)} \rho(a \mid s) u(x_s, a) \; + \; \sum_{s \in S} \pi_s \sum_{a \in \overline{A_\Delta(x_s)}} \rho(a \mid s) u(x_s, a)$$

Taking the difference between the two utilities, we get

$$\left| U(\pi, \rho') - U(\pi, \rho) \right|$$

$$\leq \left| \sum_{s \in S} \pi_s \left( \frac{1}{\rho(A_\Delta(x_s) \mid s)} - 1 \right) \sum_{a \in A_\Delta(x_s)} \rho(a \mid s) u(x_s, a) \right| + \left| \sum_{s \in S} \pi_s \sum_{a \in \overline{A_\Delta(x_s)}} \rho(a \mid s) u(x_s, a) \right|$$

$$= \left| \sum_{s \in S} \pi_s \frac{1 - \rho(A_\Delta(x_s) \mid s)}{\rho(A_\Delta(x_s) \mid s)} \sum_{a \in A_\Delta(x_s)} \rho(a \mid s) u(x_s, a) \right| + \left| \sum_{s \in S} \pi_s \sum_{a \in \overline{A_\Delta(x_s)}} \rho(a \mid s) u(x_s, a) \right|$$

$$\leq \sum_{s \in S} \pi_s \frac{1 - \rho(A_\Delta(x_s) \mid s)}{\rho(A_\Delta(x_s) \mid s)} \sum_{a \in A_\Delta(x_s)} \rho(a \mid s) \cdot B \; + \; \sum_{s \in S} \pi_s \sum_{a \in \overline{A_\Delta(\mu_s)}} \rho(a \mid s) \cdot B$$

$$= B \sum_{s \in S} \pi_s \frac{\rho(\overline{A_\Delta(x_s)} \mid s)}{\rho(A_\Delta(x_s) \mid s)} \rho(A_\Delta(x_s) \mid s) \; + \; B \sum_{s \in S} \pi_s \rho(\overline{A_\Delta(x_s)} \mid s)$$

$$= 2B \sum_{s \in S} \pi_s \rho(\overline{A_\Delta(x_s)} \mid s) \stackrel{(33)}{\leq} \frac{2B\delta}{\Delta}.$$

This proves the lemma. $\qquad \square$

We now prove Lemma F.3.

*Proof of Lemma F.3.* Consider the objective $\underline{\mathrm{OBJ}}^{\mathcal{R}}(\delta) = \sup_\pi \min_{\rho \in \mathcal{R}_\delta(\pi)} U(\pi, \rho)$. By Lemma F.4, for any $(\pi, \rho)$ there exists an agent strategy $\rho' : s \mapsto \Delta(A_\Delta(x_s))$ that only randomizes over $\Delta$-optimal actions such that $\left| U(\pi, \rho') - U(\pi, \rho) \right| \leq \frac{2B\delta}{\Delta}$. Because minimizing over $\Delta(A_\Delta(x_s))$ is equivalent to minimizing over $A_\Delta(x_s)$, which corresponds to deterministic $\Delta$-best-responding strategies, we get:

$$\underline{\mathrm{OBJ}}^{\mathcal{R}}(\delta) = \sup_\pi \min_{\rho \in \mathcal{R}_\delta(\pi)} U(\pi, \rho) \geq \sup_\pi \min_{\rho' : s \mapsto \Delta(A_\Delta(x_s))} U(\pi, \rho') - \frac{2B\delta}{\Delta}$$

$$= \sup_\pi \min_{\rho' : s \mapsto A_\Delta(x_s)} U(\pi, \rho') - \frac{2B\delta}{\Delta}$$

$$= \underline{\mathrm{OBJ}}^{\mathcal{D}}(\Delta) - \frac{2B\delta}{\Delta}.$$

$\qquad \square$

