# OpenReview forum: "Generalized Principal-Agent Problem with a Learning Agent"
_ICLR.cc/2025/Conference — ICLR 2025 Spotlight_

### Official Review · Reviewer_687t · 2024-10-30

**Soundness:** 3
**Presentation:** 3
**Contribution:** 4
**Rating:** 8
**Confidence:** 4

**Summary:**

This paper proposes a general method to reduce the repeated principal-agent games to one-shot approximate best-response on principal-agent games.

The setup is very close from the Bayesian Persuasion  one [Kamenica and Gentzkow, 2011].
The principal does not have commitment power but as in the Bayesian Persuasion setup, they can disclose information so to influence the agent’s behavior.
Then, the problem on a repeated game is reduced to a one-step game with approximately optimal action played by the agent. The reduction is achieved through the fact that a no-regret learning agent is behaving approximately optimally - which can be seen with the following equation:
$$V(\pi, d) - V(\pi, \rho) \leq CReg(T)/T,$$
which makes the agent’s actions a $\delta$-approximate best-response. After this reduction is done, the paper shows that an agent’s appropriate best response makes the principal's utility close from optimality. The proofs are achieved through a nice perturbation argument. The paper finally gives some applications to Bayesian Persuasion, Stackelberg games and Contract design.

I appreciate the fact that extending an online Bayesian Persuasion model, the paper recovers several interesting games and results. Also, the assumptions used are reasonable, which can be a lack in some papers from this kind of literature. As an example, for Theorem 3.1, the principal knows the agent’s regret bound but not the learning algorithm itself, which is a reasonable assumption and could be recovered in real-world scenarios.

**Strengths:**

I really enjoy the fact that the paper formulates interesting results about a fully abstract game theoretic setup. I think that great applications could be built relying on that.

One might wonder about the relevance of a theoretical paper on principal-agent issues for ICLR. However, the paper’s focus on signaling states and Bayesian perspectives effectively bridges representational learning with economic concepts, making it a strong fit for the conference.

The paper is well written and clear, which makes it pleasant to read.

**Weaknesses:**

**First point:** My main concern is about the relation between the paper and existing results in the computational game-theoretic literature, especially in the link between no-regret and equilibrium.

I enjoy how the paper relates online learning problem between a principal and an agent to one-step games. However, I wonder if the results that are obtained (e.g., a principal's utility arbitrary close from the optimal one, etc) can be described in terms of coarse-approximate or coarse-correlated equilibrium. More discussion on this would be interesting since the paper does not mention any form of equilibrium at all.

Regret minimization of the agents alone is sufficient to ensure convergence to coarse correlated equilibrium in general-sum multiplayer games. This kind of results are given in these lecture notes for instance (https://www.mit.edu/~gfarina/2024/6S890f24_L04_regret/L04.pdf). Is it possible to formulate your setup as a game (for instance by saying that players commit to a strategy that will be more complex and will depend on a revelation of a state of nature) and then apply the equilibrium results to obtain your convergence/approximation rates?

**Second point:** I have a few comments on the setup. According to the authors, it is a general principal-agent game with a learning agent that recovers Bayesian Persuasion among other types of games. To me, this setup is exactly an online version of Bayesian Persuasion canonical setup, adding the fact that the principal also take an action $x_t$ alongside choosing a signal. It is not a huge issue but I think that the authors should not phrase their problem as such a general setup.

Below, I copied the model of the paper « Online Bayesian Persuasion », that is part of the references:

« We consider the following online setting. The sender plays a repeated game in which, at each round $t \in [T]$, she/he commits to a signaling scheme $\phi_t$, observes a state of nature $\theta_t \sim \mu$, and she/he sends signal $s_t\sim \phi^t_{\theta_t}$ to the receiver. Then, a receiver of unknown type updates her/his prior distribution and selects an action $a_t$ maximizing her/his expected reward (in the one-shot interaction at round t). We focus on the problem in which the sequence of receiver’s types $\{k_t:t \in [T]\}$ is selected beforehand by an adversary. After the receiver plays $a_t$, the sender receives a feedback on her/his choice at round t. In the full information feedback setting, the sender observes the receiver’s type $k_t$. Therefore, the sender can compute the expected payoff for any signaling scheme she/he could have chosen other than $\phi_t$. Instead, in the partial information setting, the sender only observes the action $a_t$ played by the receiver at round t. », Castigioni et al., 2020

I agree that it is different from what is proposed in the Generalized Principal-Agent Problem with a Learning Agent setup (page 4 in the paper) but it seems quite close though. The type of the agent in Castiglioni et al. could be represented as the strategy picked by the agent at round $t$ in this model. I think that this work desserves a more thorough discussion, due to its similarity with your framework. I think that good ideas are brought to the setup but the differences should be highlighted.


I put in the weakness section my main concerns but they should be seen more as questions and I am confident in a satisfying answer to them by the authors.

**Questions:**

I would say that the paper could be defined as an attempt towards information design with a learning agent and how to manipulate/persuade this agent. To my knowledge, the same kind of issues are becoming explored recently for mechanism design or Markov Decision Processes setups with a learning agent (as Guruganesh et al., 2024 with Contract theory, that you mention):

	- Learning to Mitigate Externalities: the Coase Theorem with Hindsight Rationality, that studies mechanism design bet-ween a learning principal and a learning agent on a bandit game and how the principal can align the agent’s actions on her preferences.

	- Principal-Agent Reward Shaping in MDPs, that studies principal-agent interactions on a MDP and formulates it as an  optimization problem.

	- Bayesian Incentive-Compatible Bandit Exploration, that studies how a principal could reveal information to incentivize the agent to take some specific actions - they also provide a black-box reduction of the problem to incentive-compatible bandits..

One could say that information design is about trading information while mechanism design is about trading utility. I believe that these references are close from your work and that it could be interesting to discuss them. How could they be related it?

Finally, it is mentioned that « the sender commits to a signaling scheme » and then follows $\pi(s|\omega)$ - it is always the case in Bayesian Persuasion. However, if we consider the setup of a game, I do not see why the principal would not deviate from it if this was in their interest. I think that this part of the model should be discussed with regards to real scenarios or motivations.

---

> ### Author Response · Authors · 2024-11-24
> **Response to weaknesses**
>
> > First point: My main concern is about the relation between the paper and existing results in the link between no-regret and equilibrium. .... Is it possible to formulate your setup as a game and then apply the equilibrium results to obtain your convergence/approximation rates.
>
> We note that the existing results on the convergence of no-regret learning to equilibrium do not apply to our problem.  The existing results are: when multiple players simultaneously use no-regret learning to play a game repeatedly, they will converge to a coarse correlated equilibrium. However, our problem has only one no-regret learning player -- the agent.  The principal is not learning.  The principal aims to find the optimal strategy to play against the no-regret learning agent. And the benchmark we are comparing to is the Stackelberg equilibrium (where the principal first commits, the agent then best responds), which is different from coarse correlated equilibrium (which is a simultaneous equilibrium notion).  So, our results are not implied by the existing results in convergence of multi-agent no-regret learning.
>
> > Second point: To me, this setup is exactly an online version of Bayesian Persuasion canonical setup, .... I agree that it is different from what is proposed in the Generalized Principal-Agent Problem with a Learning Agent setup (page 4 in the paper) but it seems quite close though. The type of the agent in Castiglioni et al. could be represented as the strategy picked by the agent at round in this model. I think that this work deserves a more thorough discussion.
>
> Indeed, our model of persuasion with a learning agent is similar to online Bayesian persuasion.  However, there are two key differences between our work and [Castiglioni et al].  The first is the receiver's behavior. In our problem, the receiver does not need to know the environment or the sender's signaling scheme; instead, the receiver uses a learning algorithm to choose actions (and observes feedback after each round).  But in [Castiglioni et al], the receiver at each round knows their own utility and the sender's signaling scheme, so the receiver performs Bayesian update after receiving a signal and take the best-responding action.  The second difference is in the sender's knowledge.  In [Castiglioni et al], the sender does not know the receiver's type (hence not know the receiver's utility function); their work designs an online learning algorithm for the sender to compete against the best fixed signaling scheme in hindsight. In our work, the receiver does not have a private type, the sender knows everything including the receiver's utility function, and the sender's goal is to exploit the learning behavior of the receiver.  Given the different setups, the results from the two works are not directly comparable in our opinion.
>
> We don't think the type of the agent in [Castiglioni et al] could be represented as the strategy picked by the agent in our model, because the type in [Castiglioni et al] is adversarially chosen by the nature, while our agent's strategy is picked by the agent's no-regret learning algorithm.  Moreover, the type and the strategy affect the principal's utility in completely different ways.

---

> ### Author Response · Authors · 2024-11-24
> **Response to quesitons**
>
> > (Question 1) Relation to the references.
>
> Thank for you the references!  Despite some high-level similarities with our work, the specific models and results of those works are significantly different from ours.  In particular:
>
> - [Scheid et al 2024, Learning to Mitigte Externalities] consider a contract design setting where both the agent and the principal don't know their utility functions and use no-regret algorithms to learn. Unlike them, our principal knows the utility function and does not employ learning algorithms. Another difference is that their principal commits to the contract at each period and the agent's learning algorithm takes into account the committed contract, while our principal does not commit and the agent's algorithm is agnostic to the principal's current strategy. Although commitment is natural in the contract design setting, it is less natural in information design.  We are more interested in the information design setting and drop the commitment assumption.
>
> - In [Ben-Porat et al 2024, Principal-Agent Reward Shaping in MDPs], the principal designs the reward function of an MDP played by the agent. Crucially, the agent is assumed to be able to best respond, i.e., computing the best MDP policy, which is different from our learning agent model.
>
> - [Mansour et al, 2015, Bayesian Incentive-Compatible Bandit Exploration] study a problem where the principal, not knowing the mean rewards of different arms/actions, signals a sequence of receivers in an incentive-compatible way (namely, the receivers are willing to follow the recommendations) to learn about the arms.  We consider a principal who knows the environment trying to signal the receiver in a possibly non-IC way to maximize the principal itself's utility.  Their reduction is to show that any general (possibly non-IC) multi-armed bandit algorithm can be converted to an IC algorithm in their problem, while our reduction is to reduce the problem with a learning agent to the problem with an approximately-best-responding agent.
>
> Given the above differences, we don't see a close connection between our work and theirs.   We have added [Scheid et al 2024] to our references because their work also considers a learning agent, similar to ours, despite significant differences in the underlying models.
>
>
> > (Question 2) It is mentioned that « the sender commits to a signaling scheme » and then follows - it is always the case in Bayesian Persuasion. However, if we consider the setup of a game, I do not see why the principal would not deviate from it if this was in their interest. I think that this part of the model should be discussed with regards to real scenarios or motivations.
>
> Thank you for pointing this out!  While standard Bayesian persuasion requires the principal to commit to a signaling scheme, our model with a learning agent no longer assumes this.  We allow the principal to choose the signaling scheme $\pi^t$ _after_ the receiver chooses the signal-to-action mapping $\rho^t: S \to \Delta(A)$.  Our original model had "(1) the sender chooses a signaling scheme $\pi^t$" and "(2) the receiver chooses a strategy $\rho^t$ _simultaneously_ (based only on history, but not the sender's current round strategy $\pi^t$)".  This might cause the misunderstanding that the sender commits first.  So, we decided to switch (1) and (2) in the presentation of our model.  In the newly uploaded PDF, you will see that (1) becomes "the receiver chooses $\rho^t$ according to a learning algorithm", and (2) becomes "the sender chooses a signaling scheme $\pi^t$".  Being equivalent to the original model, this new description of our model has a clearer real-world motivation: in practice, people hardly commit to signaling schemes; receivers learn how to interpret the sender's language based on what the sender said in the past; knowing how the receiver interprets the language, the sender will then decide what to say to potentially take advantage of the receiver.
> Under this motivation, our conclusion that no-swap-regret learning agent can prevent the principal from doing better than $U^*$ is particularly interesting, because it means that, even if the principal can strategize after the agent, the principal cannot do better than the case where the principal commits first.

---

> > ### Comment · Reviewer_687t · 2024-11-25
> >
> > I thank the authors for their answers which raise many of my concerns. I believe that this paper is a great effort in a new and promising research direction. I definitely prefer the switch that the authors proposed between (1) and (2) which makes things clearer in my views. I am happy to increase my score accordingly.

---

### Official Review · Reviewer_QEC8 · 2024-11-02

**Soundness:** 4
**Presentation:** 4
**Contribution:** 4
**Rating:** 8
**Confidence:** 4

**Summary:**

The paper studies a generalized principal-agent problem where the agent learns to make decisions. The goal is to design the principal's strategy against such an agent. A sequential setting is considered, where the principal and the agent interact over time. The agent uses a no-regret algorithm to determine actions to perform in each time step, based on the past reward yielded by his actions. The paper considers different types of no-regret algorithms, including no-swap-regret and mean-based no-regret algorithms. Utilities the principal can achieve against these algorithms are analyzed, in comparison with the principal's optimal utility in the standard one-shot principal-agent model.

**Strengths:**

The paper is well-writen and very clear. I enjoy reading it. The topic of playing against a learning agent is very relevant to the theme of ICLR. Extending this line of research from standard normal-form games to generalized principal-agent problems is well motivated and interesting. The paper analyzed different types of no-regret algorithms and the results presented look quite complete. Technically, the results also look solid and are presented rigorously. The authors did a good job in explaining the intuitions behind the results, too.

**Weaknesses:**

I don't have any major concerns with the paper. One weakness is that Results 1 and 4 seem to largely follow by previous work and looks somewhat incremental. But the other results look sufficiently new and to extend normal-form games studied in previous work to generalized principal-agent problem seems to require a good amount of effort. It would be helpful if the authors can stress a bit more the differences between normal-form games and generalized principal-agent problem, and highlight the additional difficulties for addressing the latter.

**Questions:**

- I assume that the whole paper assumes the principal knows the agent's payoffs? If so it would be helpful to point that out more explicitly.

- There's also recent work on similar sequential principal-agent problems, which look relevant: Gan et al. (2023), Sequential principal-agent problems with communication: efficient computation and learning. What do you see any implications between your work and theirs?

---

> ### Author Response · Authors · 2024-11-23
>
> We thank the reviewer for the appreciation!  Regarding the differences between normal-form games and generalized principal-agent problems, two major differences are the presence of signals in the generalized problem, and a general decision space $\mathcal X$ for the principal (compared to the probability simplex over pure actions in normal-form games).  The presence of signals requires us to consider _contextual_ no-(swap-)regret algorithms, unlike the normal no-(swap-)regret algorithms in normal-form games in [Deng et al, Strategizing against no-regret learners, NeurIPS'19].  In particular, contextual no-swap-regret learning requires new analysis in the reduction from no-swap-regret learning to approximate best response (Theorem 3.4).  The general decision space $\mathcal X$ causes technical difficulty in the sensitivity analysis for approximately best responding agent (proofs of Theorem 4.1 and 4.2).
>
> > (Question 1): The principal knows the agent's payoffs?
>
> Yes, we will point this out explicitly.
>
> > (Question 2): What do you see any implications between your work and Gan et al (2023), Sequential principal-agent problems with communication: efficient computation and learning.
>
> Thanks for the reference!  Although Gan et al (2023)'s sequential principal-agent problems indeed include a large class of principal-agent problems as we do, their learning problem is significantly different from ours.  In their work, neither the principal nor the agent knows the environment, and they design a centralized algorithm (that controls both the principal and the agent) to explore the environment so as to find an optimal strategy for the principal.  In our problem, only the agent is learning, while the principal knows the environment and aims to exploit the non-best-responding learning agent.  Given the different setups, we don't see a direct connection between our and their works.

---

> > ### Comment · Reviewer_QEC8 · 2024-11-26
> >
> > Thank you for your answers. I didn't mean the work referred to subsumes your paper - just that it might be related at a conceptual level. In any case, I remain positive about your work.

---

### Official Review · Reviewer_H3BR · 2024-11-03

**Soundness:** 3
**Presentation:** 2
**Contribution:** 2
**Rating:** 5
**Confidence:** 3

**Summary:**

This paper studies "generalized principal-agent problem" in which a learning agent interacts with a principal who lacks commitment power. This setting can be used to describe several economic scenarios, like Stackelberg games and Bayesian persuasion. The paper focuses on the utility that the principal can achieve depending on the learning algorithm adopted by the agent—specifically, whether the agent uses a no-regret or no-swap-regret learning algorithm. The authors provide upper and lower bounds on the principal's achievable utility based on the algorithm played by the agent. To accomplish this, they establish a connection with the $\delta$-robust principal-agent problem, developing results that are of independent interest.

**Strengths:**

- The problem studied in the paper represents an interesting contribution to principal-agent problems that mainly focus on models in which the agent does not learn
- The results on the achievable utility when the agent plays a $\delta$-suboptimal best response according to a randomized strategy are interesting and novel

**Weaknesses:**

- If my understanding is correct, the assumption that there exists a $p_0 \ge \min \mu_0(\omega)$ limits the applicability of the results in large state instances (since ${1}/{|\Omega|} \ge p_0$), which are well studied in Bayesian persuasion problems. I believe the authors should address this limitation explicitly in the paper and discuss potential extensions.

- Similarly, in Stackelberg games with a small inducibility gap, the proposed analysis does not hold.

- The approach to proving Theorem 4.1 (first point) appears somewhat incremental with respect to Gan et al. (2023). Both consider a convex combination (depending on the inducibility gap) of two strategies, where in one strategy the principal’s utility is larger than $G$ when they play specific action compared to all the remaining actions. This is not necessarily a drawback, but it raises questions about the technical novelty of this approach—perhaps related to the shape of the domain?

- The authors show that their model can be generalized to contract design settings. However, they should more clearly specify that they consider a model without costs, particularly in comparison with prior works.

- Typo: Line 981 “Sectin”

**Questions:**

- In the statement of Theorem 5, the authors claim: "the sender can obtain a utility significantly larger than $U^*$ ..." Can the authors quantify what "significantly larger" means in the context of this theorem? Is it something proportional to the time horizon $T$? Furthermore, why does the lemma hold only in Bayesian persuasion settings?

- How can the fixed policy $\pi_t$ in the statement of Theorem 3.1 be computed?

---

> ### Author Response · Authors · 2024-11-23
> **Response to weaknesses**
>
> > (Weakness 1 and 2) The limitations regarding $p_0$ and inducibility gap.
>
> We note that the condition on $p_0$ and the inducibility gap are standard regularity conditions in the literature.  The condition on $p_0$ is necessary in robust Bayesian persuasion (as shown by Zu et al (2022), Learning to Persuade on the Fly: Robustness Against Ignorance), and the inducibility gap is well-known to be necessary in robust Stackelberg games (like Gan et al (2023), Robust Stackelberg Equilibria).  Without these two conditions, the $\delta$-approximately-best-responding strategy of the agent may significantly reduce the principal's utility -- the principal's utility is no longer lower bounded by $U^* - O(\delta)$.
>
>
> > (Weakness 3) The approach to proving Theorem 4.1 (first point) appears somewhat incremental with respect to Gan et al (2023). ... This is not necessarily a drawback, but it raises questions about the technical novelty.
>
> This part of our proof is indeed inspired by Gan et al (2023).  But our other results use different techniques.  For example, in the second point of Theorem 4.1 and Theorem 4.2, we consider randomized $\delta$-best-responding strategies for the agent, which to our knowledge have not been studied by previous works.  To our surprise, randomized strategies turn out to be qualitatively different from deterministic ones: randomized ones have $\underline{OBJ}^R(\delta) = U^* - O(\sqrt \delta)$ while deterministic ones have $\underline{OBJ}^D(\delta) = U^* - O(\delta)$.  The proof of this result is inspired by Markov's inequality (see Lemma F.3 and Appendix F.6).  We also show that this distinction between randomized and deterministic strategies is unavoidable (Example 4.1), whose proof requires analyzing a maxmin problem where the minimization step is a fractional knapsack problem.  The results and techniques regarding randomized agent strategies are new as far as we know.
>
> > (Weakness 4) The authors show that their model can be generalized to contract design settings.  However, they should more clearly specify that they consider a model without costs, particularly in comparison with prior works.
>
> Thank you for noticing this issue!  We have modified our Appendix D.2 to add costs for the agent's actions, which makes it consistent with prior works.  All our results remain unchanged.  You can see the newly uploaded PDF for details.

---

> ### Author Response · Authors · 2024-11-23
> **Response to questions**
>
> > (Question 1) Can the authors quantify what "significantly larger" means in Theorem 3.5?  ... Furthermore, why does the lemma hold only in Bayesian persuasion setting.
>
> In the proof of Theorem 3.5 (Appendix E.4), we have quantitative results: the optimal sender utility in the classic Bayesian persuasion model is $U^* = 0$, while against a $\gamma$-mean-based learning agent, the principal can obtain utility $1/2 - O(\sqrt \gamma)$, averaged over all $T$ rounds. Since $\gamma$ typically converges to $0$ as $T\to\infty$, the principal's average utility approaches $1/2$, which is larger than $U^* = 0$ by a constant.
>
> The result that the principal can exploit a mean-based learning agent not only holds in the Bayesian persuasion setting.  Previous works (Deng et al, Strategizing against no-regret learner, NeurIPS'19) showed that, against a mean-based learner, the principal can do significantly better than the Stackeberg value $U^*$ in Stackelberg games.  However, it was unclear whether this phenomenon also holds in the Bayesian persuasion setting, where the principal has private information and can send signals.  We show that the same phenomenon indeed holds in Bayesian persuasion as well.
>
> > (Question 2) How can the fixed policy $\pi_t$ in the statement of Theorem 3.1 be computed.
>
> The construction of the fixed policy $\pi_t$ in Theorem 3.1 is implicitly given in the proof of Theorem 3.1 and the proofs of Theorem 4.1 and 4.2.  In the proof of Theorem 3.1, we let $\delta=\frac{CReg(T)}{T}$ be the regret of the agent, and let $\pi_t$ be a strategy $\pi^{\varepsilon, \delta}$ that is robust against $\delta$-best-responding agent: namely, no matter what $\delta$-best-responding strategy is used by the agent, the principal can always guarantee an $\varepsilon$-optimal utility.  Then, the construction of $\pi^{\varepsilon, \delta}$ is given in the analysis for $\underline{OBJ}^R(\delta)$ in Theorem 4.1 (for unconstrainted generalized principal-agent problems) and Theorem 4.2 (constrained).  The specific construction is involved, so we didn't write it explicitly.  At a high level, it is a perturbation idea: First, compute the optimal principal strategy $\pi^*$ assuming a best-responding agent (which can be done by a linear program as shown by previous work), which is not robust because the agent might be indifferent between two actions when $\pi^*$ sends a signal; Then, perturb $\pi^*$ slightly to break indifference, so that the agent strictly prefers one action than another by $\delta$ no matter what signal is sent.  Such perturbation will decrease the principal's utility by $O(\sqrt \delta)$, hence $\pi^{\varepsilon, \delta}$ achieves $U^* - O(\sqrt \delta) - \varepsilon$ and $\pi_t$ archives $U^* - O(\sqrt \frac{CReg(T)}{T}) - \varepsilon$.

---

> > ### Comment · Reviewer_H3BR · 2024-11-27
> >
> > I thank the authors for their response. However, I still believe that the first point of the proof of Theorem 4.1 is incremental with respect to the one by Gan et al. (2024). Furthermore, I believe that the model presented in the paper does not make much sense when applied to contract design, even when considering costly actions. Indeed, in the model proposed by the authors, the agent chooses an action without knowledge of the contract the principal committed to. According to what reported by the authors in appendix D.2., the agent selects an action based on a signal, i.e., the action recommended by the principal, without knowing the actual contract the principal committed to. To me, this does not really make sense in a real-world scenario, as it would imply that an agent performs an action on behalf of the principal without knowing how much the principal will pay it.

---

> > > ### Author Response · Authors · 2024-11-27
> > >
> > > There might be a misunderstanding regarding whether the learning agent in our model knows the contract committed by the principal. What our model actually assumes is that the learning algorithm of the agent does not take into account the contract; this holds for popular no-regret learning algorithms, which choose actions based on history (not the current round information) and then adjust strategies based on the reward feedback at the end of each round. So, our results remain the same no matter whether the agent knows the principal's contract.  The motivations to consider a learning agent (even if the agent sees the contract) include, e.g., the agent does not know other parameters of the environment like the cost of each action and the transition probabilities from actions to outcomes.  More motivations can be found in previous work on contracting with a learning agent [(Guruganesh et al 2024)](https://arxiv.org/pdf/2401.16198).
> > >
> > > Second, we note that the main motivating application of our work is the information design problem, instead of contract design. In information design, the receiver not knowing the sender's signaling scheme is well motivated.  People hardly commit to signaling schemes in practice.  Receivers learn how to interpret the sender's language based on what the sender said in the past.  Knowing how the receiver interprets the language, the sender will then decide what to say to potentially take advantage of the receiver.  This is captured by our model of persuading (cheap talk) with a learning agent.

---

> > > > ### Comment · Reviewer_H3BR · 2024-11-27
> > > >
> > > > Thanks for your response. If I understand correctly, there are two possible models:
> > > >
> > > > - The agent ignores the information contained in the contract. In this case, I don’t understand why the agent would not use the information from the contract if they observe it. Indeed, if the agent does not know the parameters of the environment, they can learn these parameters and adjust their strategy based on the information gathered during the different rounds and the information contained in the contract proposed by the principal.
> > > >
> > > > - The agent does not observe the contract proposed by the principal at each round. In this case, I don’t find it convincing that the agent would take costly actions on behalf of the principal without knowing how much the principal is willing to pay to perform these actions.
> > > >
> > > > To conclude, while I understand that the main application lies in information design, I do not find the application to contract design particularly convincing. Therefore, I am willing to maintain my score.

---

> > > > > ### Comment · Reviewer_QEC8 · 2024-11-28
> > > > >
> > > > > I tend to agree with reviewer H3BR. It’s somewhat odd if the agent has to learn the contract. Just wonder if the current model fit into the case where the agent learns their cost function instead of the contract?

---

> > > > > > ### Author Response · Authors · 2024-11-28
> > > > > > **Replying to both reviewers H3BR and QEC8**
> > > > > >
> > > > > > If the agent knows the contract and learns other parameters of the environment, then the agent's learning algorithm will satisfy the following stronger notion of no-regret: for any contract, the agent has no regret (i.e., is approximately best responding) over all the periods where that contract is chosen by the principal.  We can call this notion "__contract-aware no-regret__".  Under the contract-aware no-regret notion, the problem can be directly reduced to the principal-agent problem with an approximately best-responding agent.  So, all of our main results will continue to hold (Result 1 where the principal can achieve U^* - o(1) and Result 2 where the principal cannot achieve more than U* + o(1)).  This reduction is straightforward, and hence technically not very interesting.
> > > > > >
> > > > > > So we believe that the problem with an agent who doesn't use the contract information in their learning algorithm is technically more interesting.  Not only it is studied by previous work on contracting with a learning agent [(Guruganesh et al 2024)](https://arxiv.org/pdf/2401.16198), but more importantly, it means that the agent does not have contract-aware no-regret.  The regular no-regret property means that the agent has no regret compared to taking the best fixed action in all rounds.  The best fixed action benchmark is not optimal when the principal chooses different contracts at different rounds, because the agent should respond with different actions at different rounds.  However, we show that even if the agent is not comparing against the optimal benchmark, the problem can still be reduced to the principal-agent problem with an approximately-best-responding agent, if the agent does no-swap-regret learning.  This non-trivial reduction uses specific properties of no-swap-regret, which makes the problem conceptually and technically interesting.
> > > > > >
> > > > > > In other words, the reduction from learning agent to approximately-best-responding agent holds for both contract-aware no-regret agent and (not contract-aware) no-swap-regret agent.  We choose to focus on the latter because it is technically more interesting.
> > > > > >
> > > > > > Finally, we want to emphasize again that our motivating application is information design, instead of contract design, where the agent not knowing the principal's strategy (signaling scheme) is well-motivated, and in our opinion, much more natural than the commitment assumption in classical Bayesian persuasion.

---

### Official Review · Reviewer_f1Rw · 2024-11-04

**Soundness:** 3
**Presentation:** 3
**Contribution:** 2
**Rating:** 8
**Confidence:** 2

**Summary:**

The paper considers the principal's adaptive strategy design problem against a non-regret learning agent. The paper shows that the problem can be reduced from against an approximated best response agent.  The key technique is the perturbation argument.

**Strengths:**

- The paper introduces a novel problem along with a generic solution framework. I like its results derived from a clean reductions approach.
- The paper provides the reader's sufficient knowledge about the general principal-agent problem from Gan et al. (2024).
- The paper provides many well-sketched intuitions to help us understand its proofs.

**Weaknesses:**

- The writing of the paper can be improved. For example, the paper could use a table to summarize all results and a table for all notations in this paper. While the paper is framed under the general principal-agent problem, it only discusses the Bayesian persuasion problem as its special case.
- The major drawback of this paper is that the problem itself is not well-motivated. A no-regret learning agent would assume a stationary environment, but the principal here can adaptively adjust its strategy and make the environment non-stationary. I think there is a valid problem to study in theory, but it would be much better if we can derive some practical implication from this paper's results.
- While I find the result of this paper interesting, the technical depth of this paper is relatively shallow and thus I would not recommend higher level of acceptance for this paper.

**Questions:**

Why is it necessary in this paper to drop the private information in the generalized principal-agent problem from  Myerson (1982) and Gan et al. (2024)?

---

> ### Author Response · Authors · 2024-11-23
>
> > (Weakness 1) ... While the paper is framed under the general principal-agent problem, it only discusses the Bayesian persuasion problem as its special case.
>
> While we focused on Bayesian persuasion as an example, our paper also discussed two other special cases of the general principal-agent problem -- Stackelberg games and contract design.  We provided a summary in Section 5 and a full discussion in Appendix D due to page limits.
>
> > (Weakness 2) The major drawback of this paper is that the problem itself is not well-motivated.  A no-regret learning agent would assume a stationary environment, but the principal here can adaptively adjust its strategy.
>
> We note that a no-regret learning agent does not assume a stationary environment.  A no-regret algorithm such as Multiplicative Weight ensures that the agent's actual cumulative utility is at most $O(\sqrt{T})$ worse than the cumulative utility of the best fixed action in hindsight (see, e.g., [here](https://theory.stanford.edu/~tim/f13/l/l17.pdf)), even when the environment is non-stationary.  Of course, when the environment is non-stationary, the optimal strategy for the agent in hindsight should be a sequence of possibly different actions for all time steps. But it is known that competing against the optimal action sequence is impossible (Example 1.1 of [this](https://theory.stanford.edu/~tim/f13/l/l17.pdf)), so the standard definition of (external) regret for a non-stationary environment considers a fixed action.
>
> Indeed, although is standard in the literature, competing against the best fixed action in hindsight might not seem to be the right choice in a non-stationary environment. But we want to emphasize the contrast between our results for no-swap-regret learning agents and no-regret learning agents, both of which do not compete against the optimal action sequence. We show that when the agent does no-swap-regret learning, the principal cannot exploit the agent even if the principal adjusts its strategy adaptively, while the principal can when the agent only does no-regret learning.  We think this is an interesting conceptual contribution of our work.
>
>
> > (Weakness 3) The technical depth of this paper is relatively shallow.
>
> The technique for our Result 1 (the principal can obtain $U^*-O(\sqrt \frac{Reg(T)}{T})$ against a no-regret agent) and Result 2 (the principal cannot obtain more than $U^* + O(\frac{SReg(T)}{T})$ against a no-swap-regret agent) is a reduction to generalized principal-agent problem with approximate best response, combined with a sensitivity analysis for the problem with approximate best response. The clean reduction and the generality of the analysis (which applies to all principal-agent problems) are part of our main technical contributions.
>
> The technique for Result 3 (There exists a no-swap-regret or no-regret learning algorithm under which the principal cannot do better than $U^* - \Omega(\sqrt \frac{SReg(T)}{T})$) is substantial in our opinion.  One might initially expect that the principal cannot do better than the linear term $U^* - \Omega(\frac{SReg(T)}{T})$, but we show that the correct answer is the squared-root term $U^* - \Omega(\sqrt \frac{SReg(T)}{T})$.
> Proving this result requires solving a maxmin problem (Equation (5)): maximize over the principal's strategy, minimize over the agent's $\delta$-best-responding strategy.  This maxmin problem is significantly more challenging than the maxmax problem (Equation (6)), requiring solving a fractional knapsack problem in the minimization part, together with a non-trivial analysis of the principal's utility under the solution of the fractional knapsack problem. (Details are in our Proof of Example 4.1, in Appendix F.2.)
>
> > (Question) Why is it necessary to drop the private information in the generalized principal-agent problem from Myerson (1982) and Gan et al (2024)?
>
> When the agent has private information, a no-swap-regret learning agent can be exploited by the principal (Lines 57-60 in Introduction).  One of the motivations of our work is to identify the condition under which no-swap-regret learning can prevent exploitation by the principal.  And we've found that no-private-information is a key condition.  Our paper hence focuses only on generalized principal-agent problems where the agent does not have private information.

---

> > ### Comment · Reviewer_f1Rw · 2024-11-23
> >
> > Thanks! Your response resolved many of my concerns and I have raised my rating accordingly.

---

### Meta-Review · Area_Chair_foRz · 2024-12-19

**Metareview:**

The paper studies (generalized) principal-agent problems in EconCS (including Bayesian persuasion and contract design) using a no-regret framework. I agree with the authors that the "clean reduction and the generality of the analysis" (which applies to all principal-agent problems) are a clear strength of the paper. Indeed, I find the authors' approach and refinement of existing results on whether agents can be exploited very insightful. While some concerns around contract design were raised, I felt that the authors did a good job at clarifying their position. Overall, I think the papers strengths clearly outweigh the weaknesses, and I recommend acceptance.

**Additional Comments On Reviewer Discussion:**

The reviewers were generally satisfied with the discussion. Some concerns regarding contract design were raised, and their resolution was not explicitly acknowledged by the reviewers; however, I think in the overall balance of the paper, those points were minor, and the reviewers seemed appreciative of the overall contribution and discussion.

---

### Decision · Program_Chairs · 2025-01-22

Accept (Spotlight)